# First evidence of bronze production in the Iron Age I southern Levant: A direct link to the Arabah copper polity

Tzilla Eshel [1,2]*, Yoav Bornstein[1], Gal Bermatov-Paz[1], Shay Bar[2]

**1** School of Archaeology and Maritime Cultures, University of Haifa, Haifa, Israel, **2** Zinman Institute of Archaeology, University of Haifa, Haifa, Israel

* teshel@univ.haifa.ac.il

## Abstract

This study presents new analytical data from the site of el-Ahwat, a short-lived Iron Age I settlement located at the northern edge of the Central Hill Country in Israel. The site's substantial metal assemblage, including copper and bronze spills and slag, provides direct evidence for on-site bronze production. Microstructural features indicate that primary alloying of copper and tin—rather than the re-melting of scrap—was practiced at the site. Lead isotope analysis, chemical composition, and microstructure link some of the metal specifically to the Faynan ores, and other finds to the Timna ores, suggesting that both ores, possibly controlled by a joint polity, supplied copper to el-Ahwat. These findings challenge long-standing assumptions about the localization of bronzeworking in urban lowland centers, and open new perspectives on the inland trade routes and social organization of the early Iron Age southern Levant. We propose that el-Ahwat was part of a broader and more complex network of copper distribution and bronze production, extending from the Arabah to the coast, including also peripheral highland communities.

## Introduction

The collapse of civilizations at the end of the Late Bronze Age (LBA) represents a pivotal transitional period that profoundly affected a range of interconnected polities, including the Egyptians, Hittites, Canaanites, Cypriots, Minoans, Mycenaeans, Assyrians, and Babylonians. These diverse and culturally heterogeneous societies formed part of a complex network of interaction, characterized by extensive trade and diplomatic exchange, that overcame challenges of distance, geography, and culture. The disintegration of this interlinked system was precipitated by a convergence of natural disasters and anthropogenic crises, culminating in the collapse of a highly interconnected world that encompassed imperial structures and what some have described as a form of early globalization. This transformative period marked the end of the Bronze Age and led to the decline or disappearance of major cultural

**Data availability statement:** All relevant data are within the paper.

**Funding:** This research was supported by THE ISRAEL SCIENCE FOUNDATION (grant No. 1032/23, awarded to T.E.).

**Competing interests:** The authors have declared that no competing interests exist.

traditions, including those of the Mycenaeans, Trojans, Hittites, and Babylonians [1]. The Egyptian Empire survived the upheavals of the LBA collapse, yet withdrew from the southern Levant, releasing the local Canaanite culture from their yoke [2].

Among the beneficiaries of this dramatic change were the Iron Age societies that controlled the copper production in the Arabah Valley located between modern Israel and Jordan. The rich copper deposits of Wadi Faynan in southern Jordan (Fig 1) constitute one of the best-preserved ancient mining and metallurgy districts in the world [3]. These deposits, part of the same geological formation as the smaller ore bodies at Timna on the western side of the Arabah Valley in Israel (Fig 1), were extensively exploited during the second half of the second millennium BCE, spanning the Late Bronze (evident at Timna) and Early Iron Ages [4–6]. The disruption of trade routes at the end of the Late Bronze Age, coupled with the collapse of the Cypriot copper industry, appears to have created an economic opportunity that enabled large-scale copper production in both Faynan and Timna [7].

Lead Isotope Analysis (LIA) has identified the reach of this copper trade network during the Early Iron Age, linking Faynan and Timna copper to sites in the southern Levant [10–14], Phoenicia [15], the Aegean [16], and Egypt [17,18]. However, given that the lead isotope signatures of the Faynan (DLS) and Timna (Amir/Avrona) ores overlap, the precise origin of the copper used at the destination sites is difficult to determine [11–14].

A key debate concerns the control of copper production in the Arabah during Iron Age I (ca. 1150–950 BCE), after the end of Egyptian hegemony in the region [19–21]. Central questions remain unresolved, namely: who initiated and organized this large-scale enterprise, and where were its products headed? These issues lie at the heart of discussions about technological and social change, and at the intersection of archaeology and history, particularly when archaeological data can first be linked to biblical narratives. In Faynan, these questions are especially relevant to the emergence of Iron-Age polities such as Ancient Edom and Ancient Israel, both of which may have had a vested interest in the region's copper resources.

Excavators have proposed that semi-nomadic Edomite groups engaged in herding and copper production occupied Faynan and Timna. These groups are thought to have moved seasonally across the southern Levant, northern Arabia (as indicated by Qurayyah ware ceramics; [22–24]), and even the Nile Delta [6,25–27]. Other scholars have located the seat of this early Iron Age polity in the Negev Highlands or the Beersheba Valley [28–31]. According to the latter model, Faynan copper reached Tel Masos (Fig 1), a site envisioned as a "gateway community" that distributed goods to the Mediterranean coast and Egypt [31–35].

The identity of the polity organizing copper production and trade has been disputed, as well. Whereas some scholars have suggested an Edomite polity, either situated in the Negev highlands [28,29] or nomads who left no significant architecture [36], others have regarded the evidence as an indication for the presence of early Israelites, either emerging from a settled, local, Canaanite population [37,38] or of nomadic origin, possibly settling in the Central Hill Country [39].

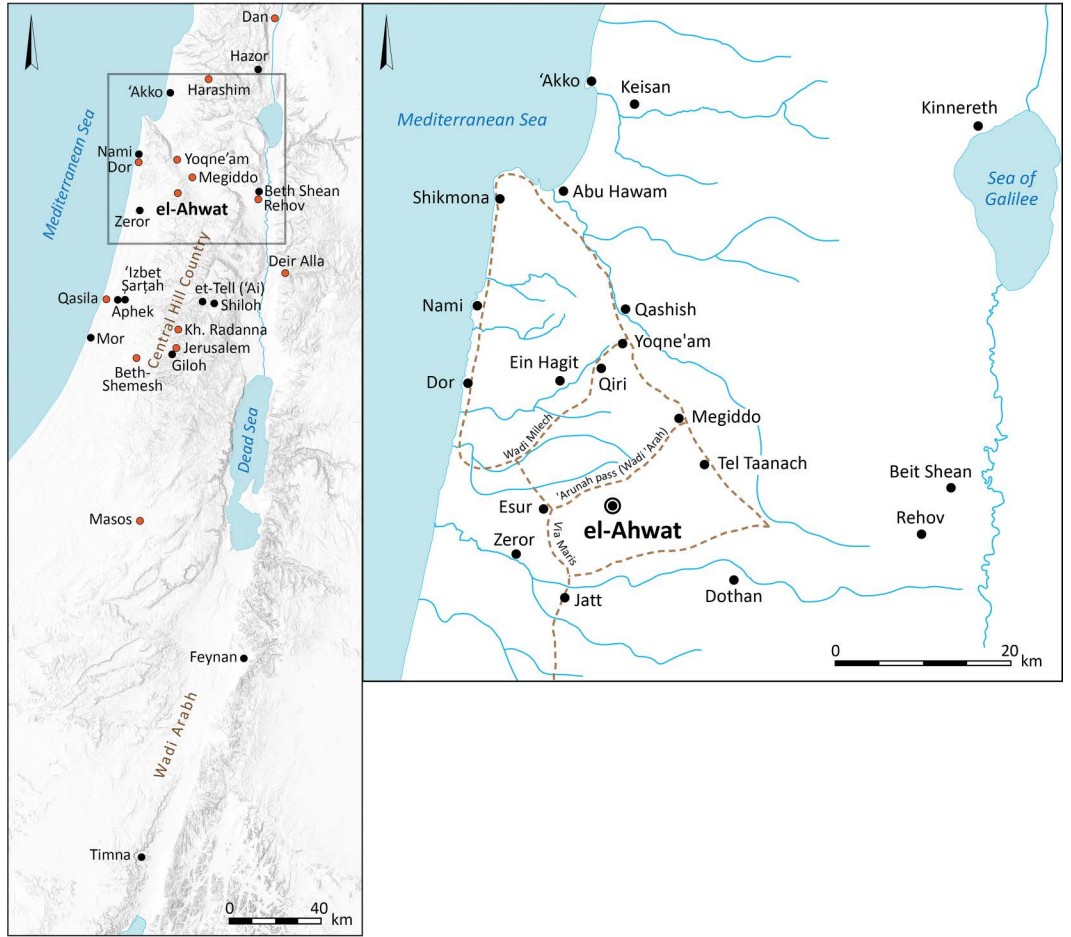

**Fig 1. a. Map of the southern Levant with sites mentioned in the text. b. Map of el-Ahwat and vicinity, including nearby sites populated in the Late Bronze Age and/or Iron Age I and main roads.** Sites with evidence of Iron Age I metalworking are marked in red. Maps prepared by Sapir Haad; reconstruction of ancient roads after [8]: Fig. 41.1; [9]: Fig. 4.17.

This debate often overlooks evidence from urban centers of the southern Levant. Although many were heavily damaged in the Late Bronze–Iron Age transition (e.g., Megiddo, Beth-Shean, Tel Reḥov, Gezer), several were rebuilt within decades and reached a new urban zenith [40]: 367–374, 472, [41–43]:107–108, [44]. These rebuilt centers became increasingly engaged in regional trade with Egypt, Cyprus, and the Levantine coast [42,45–50]. Bronzeworking was particularly prominent in the Iron Age IB, with evidence from fifteen urban sites, including Tel Dan, Tel Harashim, Tel Dor, Tel Megiddo, Tel Rehov, Tell Deir 'Alla, Tel Yoqne'am, Khirbet Raddana, Jerusalem, Tel Qasile, Tel Beth-Shemesh, and Tel Masos (Fig 1) [11–13,51–54]. Some of these settlements show continuity from the Late Bronze Age (e.g., Hazor, Megiddo, Dor), while others present no evidence of prior metalworking.

Bronzeworking during Iron Age I has been described as a decentralized, local form of production, in contrast to the centralized iron industries of Iron Age IIA [55]. Although crucibles, prills, and sediment residues have been chemically and isotopically analyzed at many sites, it remains unclear whether artisans were primarily re-melting scrap or alloying copper with tin (or tin oxide). Most scholars have assumed that re-melting was the dominant practice, typically carried out at the household level [12,53].

Despite the ample evidence of bronzeworking, the trade routes that connected Arabah copper with these urban centers have rarely been discussed, and the significance and role of copper in the local economy has been questioned [30]. Most reconstructions have focused on the route through Tel Masos and the Beersheba Valley to the coast or directly to Egypt [30]. Another possible route traversed the Transjordanian highlands (ancient Moab) along the King's Highway, a corridor evidenced in later periods [47,56–59]. Although some scholars have argued that Moab was not well integrated into the interregional exchange systems during the Iron I–IIA [60], inland trade routes within the southern Levant have rarely been considered.

Even less attention has been paid to bronzeworking in the Central Hill Country, where hundreds of new, small, often unfortified villages appeared in Iron Age I. These sites typically lack evidence of settlement hierarchy, monumental architecture, or luxury goods [61–65], and only a handful shows traces of planning (e.g., Shiloh, Giloh, et-Tell ['Ai], 'Izbet Ṣarṭah, and el-Ahwat; [66,67]). Bronzeworking has customarily been viewed as a lowland activity [61] (for a dissenting view, see [68]:100, and below). As a result, the ongoing debate regarding the origin of the groups that eventually established the northern Kingdom of Israel and the southern Kingdom of Judah during the 10th or 9th centuries BCE, has been isolated from the political and economic context of copper production in the Arabah in the preceding centuries (e.g., [30]).

In this context, the site of el-Ahwat (Fig 2) is exceptional. A short-lived Iron Age I settlement located at the northern edge of the Central Hill Country, it is notable for its large area, unique architecture, and peripheral location [69,70]. After eight excavation seasons, it became evident that the site's glyptic assemblage shows strong affinities to Egypt [69], while its substantial metal assemblage—tools, jewelry, and casting spills—reflects typical Iron Age I local styles, and was produced using Arabah copper [67]. An installation identified by the excavators as an iron furnace (U407; [71]) supposedly pre-dates all other evidence of iron smithing in the Levant, but the identification was based solely on the presence of iron minerals in the nearby soil rather than iron blooms or slag, which are typically discarded near an iron workshop ( [67] and references therein).

The site, however, did provide evidence of bronzeworking, which is analysed and presented below for the first time. Renewed excavations onsite starting 2024 (Fig 2) aim to answer unresolved archaeological questions regarding the site,

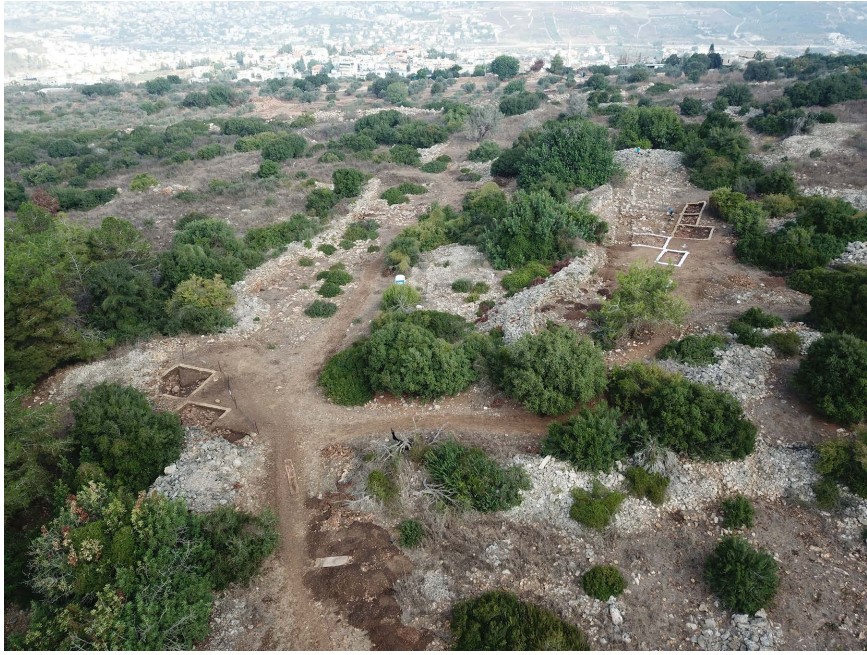

**Fig 2. El Ahwat, looking north, during excavation season Sep. 2024.** Photograph by Aaron Lipkin.

including the dating of its surrounding walls, its possible connection to the Sea People, its precise chronology within Iron Age I and the location of the bronzeworking activities within the site [67].

Fourteen metal fragments and one slag fragment were found in various locations during the eight seasons of Zertal's excavations of el-Ahwat. These finds are rare and unique, as such a large quantity of bronzeworking fragments has not been reported in other Iron Age sites in the southern Levant. Of these, seven copper and bronze fragments and the slag fragment were analyzed for microstructure, chemical composition and lead isotopes. The results, presented below, indicate that bronze production took place on-site, thereby offering a new perspective on Iron Age I copper trade and social organization in the southern Levant.

## Materials and methods

Seven metal fragments and one slag fragment were analyzed in this study (Table 1; Fig 3). The fragments generally look like irregular blobs of metal, seemingly the solidified residual deposits of copper. They are described below as 'spills'. The one exception is fragment AM_1, a relatively large, thick and structured piece of copper, which may therefore have been a cut ingot.

### Optical microscopy and SEM–EDS (Scanning Electron Microscopy–Energy Dispersive Spectroscopy)

Optical microscopy (OM) was employed to generate an overview of the microstructure of the metals and slag. Samples were prepared according to standard metallographic procedures. The samples were sectioned using an abrasive cut-off wheel on a QATM Qcut 150 saw. The sections were mounted in epoxy resin and polished down to 1 µm using a Struers Laboforce-Mi polisher. The metallographic samples were etched using Alcoholic Ferric Chloride. A Leica DM-2700 M optical microscope equipped with a polarizer was used to document and map the mounted objects.

Targeted SEM-EDS analysis was used to detect the composition of the metals and adhering slag. Analyses were performed using a Thermo Scientific Phenom XL G2 desktop SEM equipped with an EDS system. Images were acquired in Back Scattered Electrons (BSE) mode, and elemental analysis was performed at 10–15 kV and 7.8–8.9 mm working distance.

### Detailed chemical analysis (ICP-MS)

The copper and bronze spills and a copper prill within the slag fragment were further subjected to detailed chemical and Pb-isotopic analyses (see below). Sample preparation and analysis were performed in the Metals and Materials Laboratory of the School of Archaeology and Maritime Cultures at the University of Haifa. The items were drilled using a 1 mm drill. To avoid external contamination, surface drillings were discarded. Subsequently, 20–25 mg of the drillings were dissolved in $HNO_3$ and diluted with 10 mL of distilled water. The samples were analysed at the Israel Geological Survey by Dr. Nadiya Teutsch for Cu, Al, Fe, Ca, Mg, Na, K, S, Sn, Pb, As, Ag, Co, Mn, Ni, Sb, Zn, U and Th using ICP-MS (Agilent 7500cx) after calibration with external multi-element standards (Merck ME VI). The possible effect of metals from the acid used in the procedures was

**Table 1. The metal and slag fragments analyzed in this study.**

| Artefact | Figure | Description | Locus | Basket | Area | Context |
|---|---|---|---|---|---|---|
| AM_1 | 3a | copper ingot | 6408 | 64028 | D | floor U422 |
| AM_2 | 3b | copper spill | 3130 | 31131 | A | |
| AM_3 | 3c | copper spill | 7409 | 74255 | D | fill U430 |
| AM_4 | 3d | copper spill | 5408 | 54057 | D | |
| AM_5 | 3e | bronze spill | 6105 | 61080/1 | A3 | complex 100 |
| AM_6 | 3f | bronze spill | 1311 | 23169 | C1 | floor U303 |
| AM_7 | 3g | bronze spill | 5429 | 54249 | D | floor U419 |
| AM_8 | 3h | slag fragment | 1211 | 121116 | B | section in wall, mixed context |

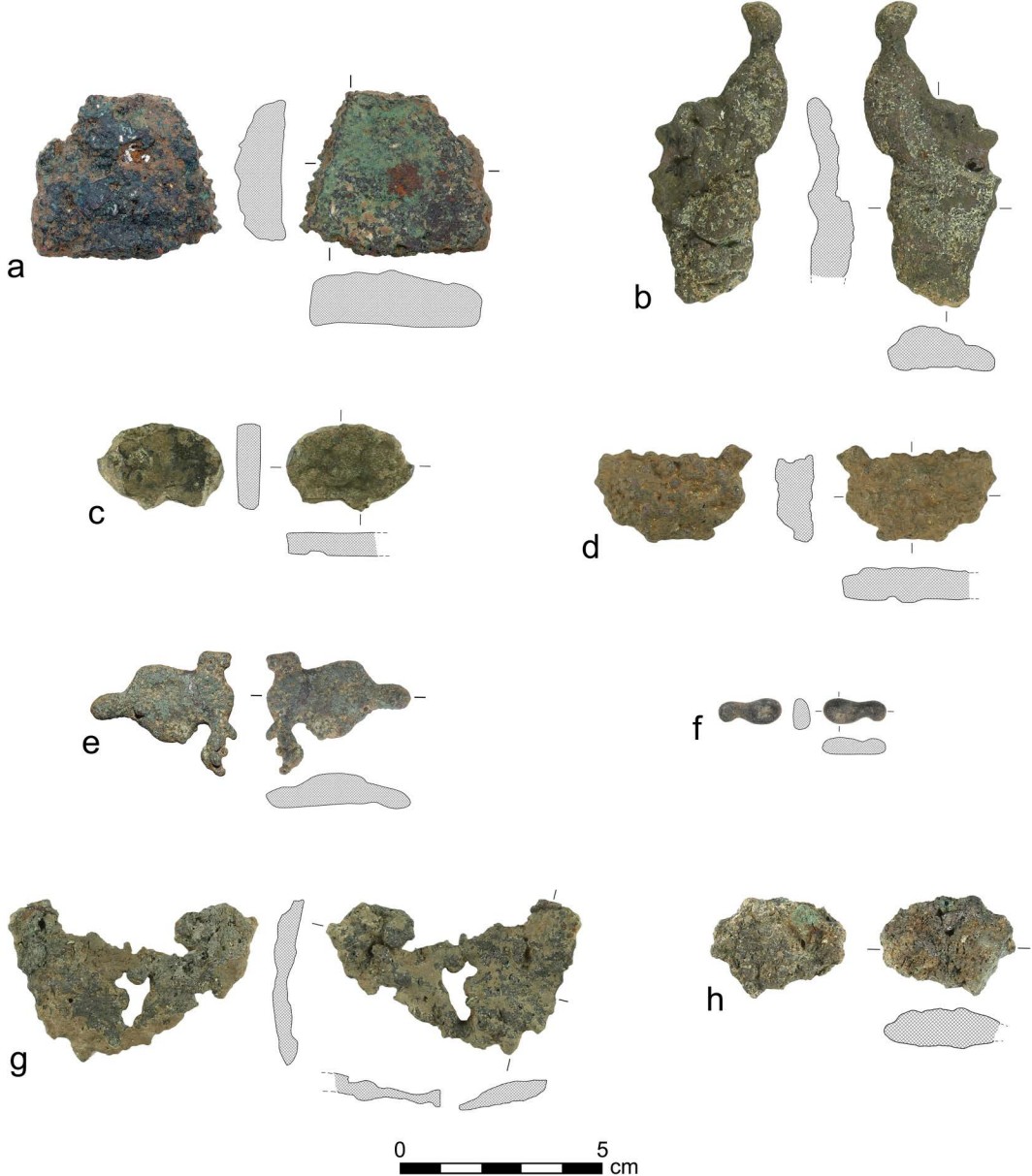

**Fig 3. The copper and bronze spills analyzed in this study.**

monitored by measuring procedural blank samples. In addition, standard reference samples (USGS SRS T-183, T-175) were examined at both the end of the calibration and at the end of the analysis for precision accuracy and detection limit estimation. The blanks were always lower than 1%. The precision and accuracy of the ICP-MS were ±5% for all samples.

## Lead Isotope Analysis (MC-ICP-MS)

Pb-isotopic ratios were used to identify copper ore sources. For a full explanation of the methodology and its limitations as well as the graphic representation of the two-stage geological model ages, on which the results are plotted, see [72]. Column separation and isotopic measurements were conducted at the Israel Geological Survey by Dr. Nadiya Teutsch

using an MC-ICP-MS (Thermo Neptune Plus) mass spectrometer. Mass fractionation corrections for Pb in both samples and the NIST-981 standard were based on a $^{205}$Tl/$^{203}$Tl isotopic ratio of 2.3875 after adding a Tl solution (50 ppb) to both the NIST 981 standard and the samples. Replicate measurements of National Institute of Standards and Technology (NIST) SRM-981 standards yielded mean values of $^{206}$Pb/$^{204}$Pb = 16.9383 ± 0.0051, $^{207}$Pb/$^{204}$Pb = 15.4939 ± 0.0059 and $^{208}$Pb/$^{204}$Pb = 36.7035 ± 0.0180 (2σ, n = 12).

## Results

### Optical microscopy and SEM-EDS

Analysis reveals two types of metal spills: copper and bronze, and one slag with evidence of bronze production.The SEM-EDS compositions of the different areas in the samples are presented in Table 2.

**The copper spills.** Four copper spills were identified (AM_1, AM_2, AM_3, AM_4; Figs 4–7), one of which (AM_1) may also be described as a cut ingot (see above). Their SEM-EDS chemical compositions are presented in Table 2 above. In all four spills, lead inclusions are evident, indicating that the copper ore contained lead, which is insoluble in copper. Spherical inclusions indicate that the spills were not cold-worked post-production.

In AM_1 and AM_2 (Figs 4 and 5), Fe is incorporated within the matrix (Specs A1, B1). Iron probably entered the copper during smelting, either due to low temperatures (below 1094°C) or a strong reducing atmosphere, where it could not be fully oxidized or removed from the metallic phase. The presence of iron made the metal harder and more brittle.

Microstructural analysis of AM_1, AM_2 and AM_3 (Figs 4-6) reveals that they are interspersed with a large number of copper-sulfide globules, indicating an incomplete separation of metal and slag during the smelting or re-melting process. This suggests a low-temperature process or a short smelting duration, which did not allow the complete separation of the molten copper from the slag, and limited the ability of sulfides to aggregate into larger particles. In contrast, in AM_4 (Fig 7) the copper-sulphide globules (dark grey) and a few Pb (white) and Fe (black) inclusions are concentrated mainly in grain boundaries. Solidification shrinkages are also seen (Fig 7a). All this indicates a more refined process, including high temperatures and slow cooling, which allow more time for impurities to segregate to grain boundaries, reducing impurity concentration within the grains.

Microstructural analysis of AM_2 and AM_3 (Figs 5 and 6) reveals Pb-rich (specs B5, B8 and C2), Mn-rich (specs B4, C3 and C4) and Fe-P-rich (iron phosphides, Specs B2, B6, B7 and C5) globules within the sulfur. In AM_3, Co-rich iron phosphides were detected (spec C5). Slag was identified trapped in copper spill AM_2 (Figs 5f-g; specs B9–B12), and adhering to the copper in AM_3. Significantly, dolomite crystals are evident within the AM_3 slag (Fig 6e; Spec C6). These have important implications regarding the source of the copper (see below).

**The bronze spills.** Optical Microscopy and SEM images of AM_5, AM_6 and AM_7 reveal that they are bronze spills. Their chemical composition is detailed in Table 2 above.

The microstructure of AM_5 (Fig 8) and AM_6 (Fig 9) is similar. Both consist of dendrites, typical of a slow cooling process during solidification. The dendrites consist primarily of pure copper (Cu), while the interdendritic material contains copper-tin (Cu-Sn) precipitates, reflecting that the tin fully segregated to the grain boundaries as the alloy cooled. The presence of relatively large, distinct globules of Cu-S (dark grey), Cu-Sn (light grey), Pb (white), and S-Fe-O (black) within the interdendritic material of both samples, further indicates that the impurities originated from the ore and that the cooling process was slow enough to enable the precipitation of these elements into inclusions. The above-described dendritic structure indicates that here, too, the bronze was not subjected to mechanical working after casting. Taken together, these observations suggest that the bronze underwent slow cooling without any further deformation.

The dendrites of AM_5 appear relatively uniform in size. The grain boundaries also contain Mn (spec E2) and As (~1–2 at. %; specs E3 and E4). In comparison, the dendrites of AM_6 are smaller, suggesting a faster cooling rate than for AM_5. In addition, the inclusions in the interdendritic material of AM_6 are smaller than in AM_5. This may be yet another result of a more rapid cooling rate, wherein impurities are less likely to coalesce into larger inclusions, resulting in smaller, more dispersed particles. Alternatively, it may be due to a more refined smelting process (higher temperatures or better

**Table 2. SEM-EDS Chemical composition of phases identified in the samples of metal spills and slag [at. %].**

| Spec | Sample | Description | O | Al | Mg | Si | Ti | P | S | Mn | Co | Fe | Ca | Cu | Zn | Ni | Na | As | Sn | Pb |
|---|---|---|---|---|---|---|---|---|---|---|---|---|---|---|---|---|---|---|---|---|
| A1 | AM_1 | Fe-rich Cu grain | 1.2 | | | | | | | | | 5.5 | | 92.0 | | | | 0.1 | 1.1 | 0.1 |
| A2 | AM_1 | Cu-S globular inclusion | 2.5 | | | | | | 24.7 | | | 11.8 | | 58.6 | | | | | 0.7 | 1.7 |
| A3 | AM_1 | Pb globular inclusion | 21.3 | | | | | | | | | 3.3 | | 13.2 | | | | 0.4 | | 61.5 |
| B1 | AM_2 | Cu grain matrix | | | | | | | | | | 1.3 | | 98.8 | | | | | | |
| B2 | AM_2 | Fe-P globular inclusion (with Co) | 0.3 | | | | | 24.2 | 0.3 | | 2.4 | 68.0 | | 4.7 | | | | | | |
| B3 | AM_2 | Cu-S globular inclusion | | | | | | | 34.8 | 0.3 | | 9.2 | | 52.3 | | | | | | 0.6 |
| B4 | AM_2 | S-Mn-Fe inclusion | 4.9 | | | | | | 44.8 | 35.3 | | 11.5 | | 3.4 | | | | | | |
| B5 | AM_2 | Pb globular inclusion | 22.4 | | | | | | | 0.2 | | 2.1 | | 36.0 | | | | 0.3 | | 38.8 |
| B6 | AM_2 | Fe-P inclusion | 6.0 | | | | | 5.1 | 3.8 | | | 18.0 | | 64.3 | | | | 0.1 | | 2.8 |
| B7 | AM_2 | Fe-P inclusion (in S) | 3.8 | | | | | 2.3 | 22.3 | | | 5.7 | | 64.6 | | | | | | 1.3 |
| B8 | AM_2 | S-Fe-Pb inclusion | 1.3 | | | | | | 15.2 | | | 0.6 | | 75.7 | | | | | | 6.8 |
| B9 | AM_2 | slag | 54.1 | 5.9 | | 6.7 | | | | 0.1 | | 0.5 | | 32.2 | | | | | | 0.3 |
| B10 | AM_2 | slag | 38.4 | 4.4 | | 12.4 | | | 0.5 | | | 33.9 | | 10.0 | | | | | | |
| B11 | AM_2 | slag | 60.5 | 9.6 | | 15.6 | | | 1.2 | 1.1 | | 4.9 | | 6.3 | 0.6 | | | | | |
| B12 | AM_2 | slag | 11.2 | | | 1.4 | | | 28.8 | | | 0.9 | | 56.8 | | | | | | 0.8 |
| C1 | AM_3 | Cu grain matrix | | | | | | | | | | | | 99.9 | | | | | | |
| C2 | AM_3 | Pb inclusion | 20.7 | | | | | | | | | | | 15.9 | | | | | | 59.5 |
| C3 | AM_3 | S-Mn-Fe inclusion | | | | | | | 47.5 | 38.8 | | 9.8 | | 3.2 | | | | | 0.1 | 0.7 |
| C4 | AM_3 | S-Mn-Fe inclusion | | | | | | | 35.3 | 0.5 | | 5.5 | | 55.5 | | | | | | 0.4 |
| C5 | AM_3 | Fe-P inclusion (with Co) | | | | | | 18.4 | | | 10.1 | 57.0 | | 13.7 | | | | | | |
| C6 | AM_3 | dolomite mineral within slag | 69.1 | | 10.6 | 3.4 | | | | | | | 11.5 | 4.9 | | | | | | |
| C7 | AM_3 | slag | 59.8 | 0.5 | | 21.7 | | | | | | | | 17.6 | | | | | | |
| C8 | AM_3 | slag | 60.4 | | | 22.0 | | | | | | | | 16.9 | | | | | 0.2 | |
| C9 | AM_3 | Ti-rich inclusion within the slag | 57.5 | 0.9 | 1.0 | 5.7 | 10.1 | | | | | 16.2 | | 8.2 | | | | | 0.3 | |
| D1 | AM_4 | Cu grain matrix | | | | | | | | | | | | 100.0 | | | | | | |
| D2 | AM_4 | Cu-S globular inclusion | 1.3 | | | | | | 31.7 | | | | | 66.9 | | | | | | |
| D3 | AM_4 | S-Pb inclusion | 18.2 | | | | | | 22.9 | | | | | 28.4 | | | | | | 30.1 |
| E1 | AM_5 | Cu dendrite | | | | | | | | | | | | 98.3 | | | | | | |
| E2 | AM_5 | Cu-Sn interdendritic material (with Mn and As) | | | | | | | | 3.3 | | 9.6 | | 75.3 | | | | 1.3 | 8.8 | |
| E3 | AM_5 | S-Fe inclusion within interdendritic material | 2.7 | | | | | | 26.2 | | | 12.3 | | 57.4 | | | | | | 1.2 |
| E4 | AM_5 | Cu-Sn inclusion within interdendritic material (with As) | 5.0 | | | | | | | | | | | 73.9 | | | | 1.5 | 19.5 | |
| E5 | AM_5 | S-Fe inclusion within interdendritic material | 23.0 | | | | | | 7.9 | | | 13.2 | | 54.6 | | | | | | 0.8 |
| F1 | AM_6 | Cu dendrite | 1.2 | | | | | | | | | | | 98.4 | | | | | | |
| F2 | AM_6 | Cu-Sn g interdendritic material | 2.1 | | | | | | | | | | | 90.6 | | | | | 6.8 | |
| F3 | AM_6 | Cu-S inclusion within interdendritic material | 1.8 | | | | | | 28.1 | | | | | 68.1 | | | | | 0.5 | 1.2 |
| F4 | AM_6 | Sn inclusion | 6.4 | | | | | | 0.2 | | | | | 70.5 | | | | | 22.5 | |
| F5 | AM_6 | S-Pb inclusion (with Sn) | 9.6 | | | | | | 8.7 | | | | | 51.4 | | | | | 1.7 | 27.4 |

*(Continued)*

 

| Spec | Sample | Description | O | Al | Mg | Si | Ti | P | S | Mn | Co | Fe | Ca | Cu | Zn | Ni | Na | As | Sn | Pb |
|------|--------|-------------|---|----|----|----|----|---|---|----|----|----|----|----|----|----|----|----|----|----|
| G1 | AM_7 | Cu dendrite (with Fe, Sn and Mn) | | | | | | | | 3.8 | | 9.8 | | 80.0 | | | | | 2.3 | |
| G2 | AM_7 | Cu-Sn interdendritic material (with Fe and Mn) | | | | | | | | 3.0 | | 8.7 | | 81.9 | | | | | 5.5 | |
| G3 | AM_7 | Cu-S inclusion within interdendritic material | | | | | | | 27.5 | | | | | 70.5 | | | | | | 0.6 |
| G4 | AM_7 | Pb-Cu-S inclusion within interdendritic material | 5.9 | | | | | | 18.6 | | | | | 63.7 | | | | | | 11.9 |
| G5 | AM_7 | Sn inclusion in slag | 74.8 | | | | | | | | | 0.6 | | 0.4 | | | | | 24.2 | |
| G6 | AM_7 | slag | 54.6 | 3.3 | 6.6 | 0.6 | | | | | 0.9 | 30.8 | 0.9 | 0.8 | | 1.1 | | | | |
| G7 | AM_7 | slag | 69.1 | 1.8 | | 6.0 | | | | | | 4.7 | 9.7 | 0.3 | | | | | 8.4 | |
| H1 | AM_8 | Cu prill | 2.4 | | | | | | | | | | | 96.0 | | | 1.5 | | | |
| H2 | AM_8 | Sn oxide | 72.2 | | | | | | | | | | | | | | | | 27.6 | |
| H3 | AM_8 | Si-rich slag (with P and As) | 61.9 | 3.6 | 3.2 | 16.1 | | 1.2 | | | | | 3.1 | 8.9 | | | | 1.4 | | |
| H4 | AM_8 | Si-rich slag (with P) | 59.0 | 2.7 | | 14.5 | | 0.8 | | 0.6 | | 5.1 | 1.4 | 10.0 | | | | | 5.7 | 0.1 |
| H5 | AM_8 | Fe oxide (with Mn) | 58.3 | 0.9 | | | | | | 1.1 | | 38.1 | 0.4 | | | | | | 1.0 | |
| H6 | AM_8 | Sn oxide (with Mn and Fe) | 67.1 | | | | | | | 1.2 | | 6.6 | 1.6 | | | | 3.5 | | 19.8 | |
| H7 | AM_8 | Sn oxide | 34.7 | | | | | | | | | 1.3 | | 1.0 | | | | | 62.8 | |
| H8 | AM_8 | Sn-Cu prill (with As) | 66.2 | | | 1.2 | | | | | | | | 12.4 | | | | 4.2 | 15.4 | |
| H9 | AM_8 | Sn-Cu prill (with As) | 17.9 | | | | | | | | | | | 59.4 | | | | 2.5 | 20.2 | |

flux use), or a subsequent purification process, which may have allowed for a more thorough separation of slag from the molten metal, resulting in fewer impurities.

Sample AM_7 exhibits a mixed microstructure, comprising both dendritic and equiaxed grains, consistent with non-uniform cooling during solidification (Fig 10). This pattern typically forms when the exterior of the metal cools rapidly, promoting dendritic growth, while the interior cools more slowly, allowing the development of equiaxed grains.

Elemental analysis of sample AM_7 shows that tin is partly dissolved within the dendrites and partly segregated to the interdendritic regions, as expected in tin bronze solidification. In this sample, the segregation of Mn, Sn and Fe to the interdendritic region is non-uniform (Specs G1 and G2). As Fe is virtually insoluble in Cu, it is unlikely to be truly incorporated into the Cu matrix. Therfore, its presence may reflect fine Fe-rich inclusions or complex phases involving Sn and Mn. This elemental distribution likely reflects the variable cooling rates across the sample: rapid cooling in some of its parts inhibited the diffusion of alloying elements and impurities, leading to their retention within the dendritic cores, while slower cooling in other areas allowed partial segregation to the interdendritic areas.

The presence of Cu-S globules (dark grey), concentrated primarily within the interdendritic solid, further supports a high-temperature smelting process followed by relatively slow cooling, in at least parts of the object. Given that sulfide segregation requires sufficient thermal energy and time to occur, its localization in the interdendritic areas suggests controlled cooling conditions consistent with refined metallurgical practice. Taken together, these observations point to a complex thermal history, involving both rapid and gradual cooling phases, possibly reflecting variation in object thickness or surface exposure during solidification.

The piece of slag adhering to the AM_7 bronze spill was found to be rich in tin (specs G5 and G7). Particles of tin oxide ($SnO_2$, or cassiterite) were detected (Fig 10f). Tin-rich slag inclusions have often been regarded as immiscible in solid

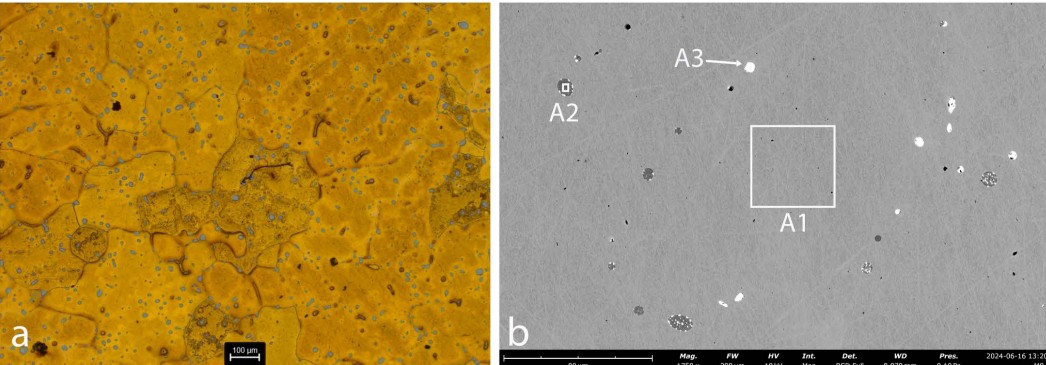

**Fig 4. Sample AM_1. a. OM, etched. b. SEM.** Cu grains containing Fe are evident (spec A1). The matrix contains a large number of globular inclusions of copper-sulfide (grey) (spec A2). Pb globular inclusions were also identified (white) (spec A3).

Cu, and interpreted as remnants of tin ore (e.g., [73]). However, experiments have shown that they can also form during simple bronze melting under oxidizing conditions [74], and therefore are not, in themselves, diagnostic of primary alloying. Nonetheless, the chain of tiny, blocky tin oxide crystals visible in Fig 10f resembles the morphology of natural cassiterite, suggesting that this spill likely resulted from the alloying of tin and copper (cf. [74]: Fig 8).

**The slag.** AM_8 is a slag fragment. Its chemical composition is presented in Table 2 above.

Optical and SEM images (Fig 11) reveal that the slag consists of a glassy Si-rich matrix (dark grey; specs H3, H4) containing residual Cu, inclusions of Fe oxide (medium-dark grey, spec H5), Sn oxide (re-crystalized cassiterite, white-speck H7 and medium-light grey-spec H6), some P, Mn and As contaminations and Cu-Sn prills (white, specs H8, H9). Since Sn-Cu prills can only be created in the process of alloying copper with tin [75], the presence of such a prill indicates that this slag is the remains of a bronze production process rather than bronze recycling.

## Detailed chemical analysis (ICP-MS)

The results of the detailed chemical analysis are presented in Table 3.

The ICP-MS results (Table 3) are compatible with the SEM-EDS results (Table 2). The chemical composition of AM-8 is mainly of a copper prill within the slag, although some of the slag material (Al, Fe, Ca, Mg, Na, K) was clearly dissolved with the prill (Table 3: AM_8). No correlation has been detected between Sn concentrations and other elements, suggesting that pure tin was alloyed with the copper. A direct proportionality between Mn and Co, measured in the copper and bronze (Fig 12), indicates that the presence of one is associated with the other.

## Lead Isotope Analysis (MC-ICP-MS)

All items were subjected to lead isotope analysis (Table 4). The results are consistent with the local Arabah ores (Timna and DLS ores in Faynan; Fig 13) and are similar isotopically to two copper earrings from el-Ahwat [67].

Copper from el-Ahwat (yellow icons in Fig 13) differs in isotopic composition from bronze (red icons in Fig 13), generally containing lead from an older geological origin. This observation suggests that lead from a younger geological origin was added to the alloy with the tin.

Notably, the Faynan ores are isotopically more condensed than the Timna ores [5,76], and only one of the items (AM_1) is consistent isotopically with Faynan DLS ores. The remaining results fall in the more-dispersed results associated with the Timna sandstone. The results are also consistent with the copper ores at Sa Duchessa in SW Sardinia,

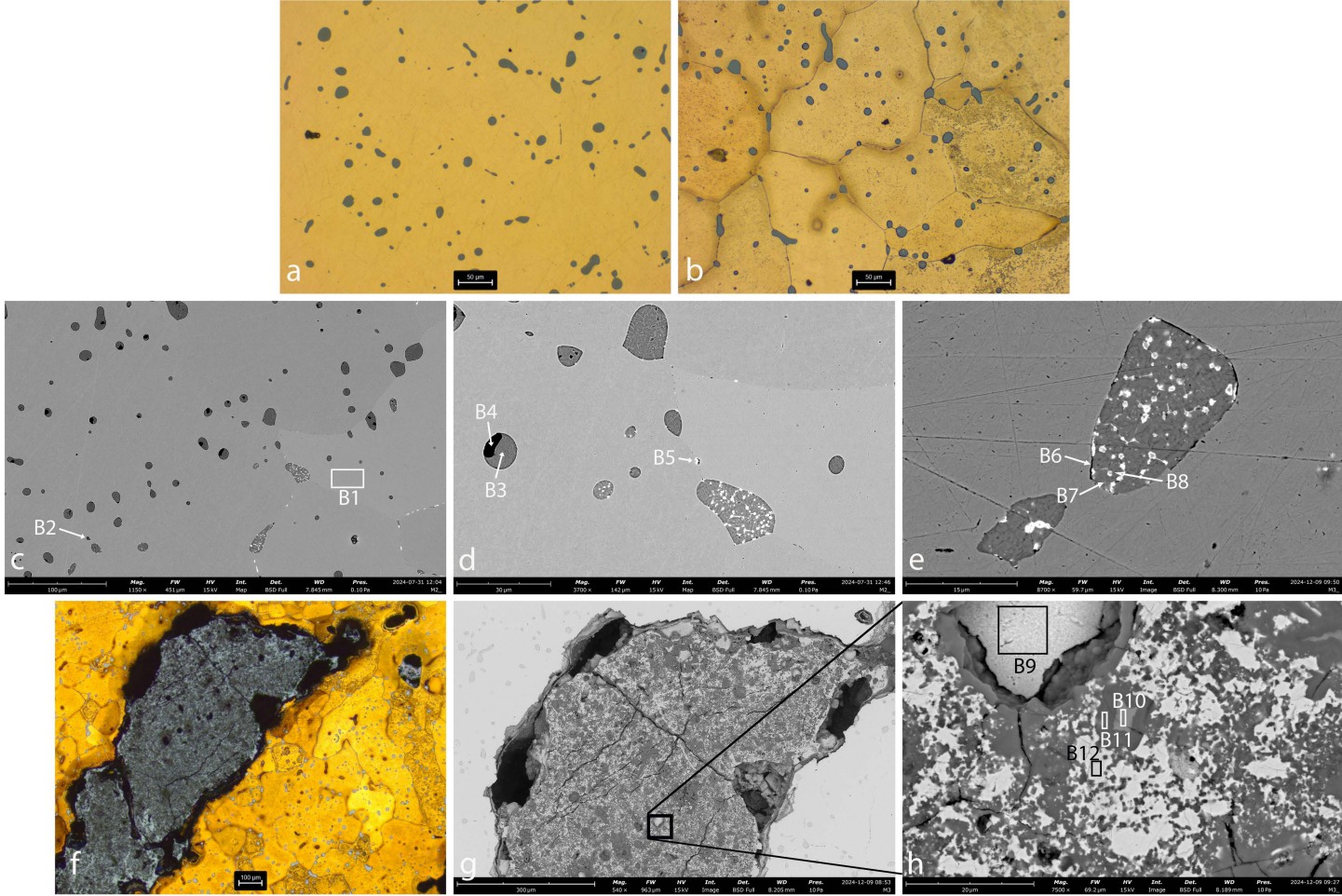

**Fig 5. Sample AM_2. a. matrix, OM, unetched. b. matrix, OM, etched. c-e. matrix, SEM. f. slag inclusion, OM, etched. g-h. slag inclusion, SEM.**
The matrix revealed large Cu grains (spec B1) interspersed with globules of copper-sulfide of varied sizes (spec B3; dark-grey). Within the sulfur, Pb-rich (spec B5; white) and Mn-Fe-sulphide (spec B4; black) globules are evident. Iron is present in the copper. High P concentrations were measured in several S-Fe globules (specs B2, B6, B7). A slag inclusion entrapped in the copper matrix contains Si, Cu, Al, S, Mn, Fe and Pb. Cu and Fe-oxides are evident in the slag (specs B9–B12).

however the Arabah ores are a more probable source, as they are local, and were the main suppliers of copper to the southern Levant during Iron Age I ([67] and additional references below).

## Discussion

### Bronzeworking at el-Ahwat: First evidence of South-Levantine bronze production

The analysis of the copper and bronze spills, along with the slag samples, indicates that bronze was produced at el-Ahwat during Iron Age I. The metallographic examination revealed that all copper and bronze items are unworked metal spills, containing inclusions of copper sulfide, lead and iron. However, the quality of the metal spills was not uniform across the assemblage. Only one copper spill (AM_4) appears to have undergone a refined smelting process, as evidenced by high smelting temperatures and slow cooling. These conditions allowed impurities to segregate toward the grain boundaries,

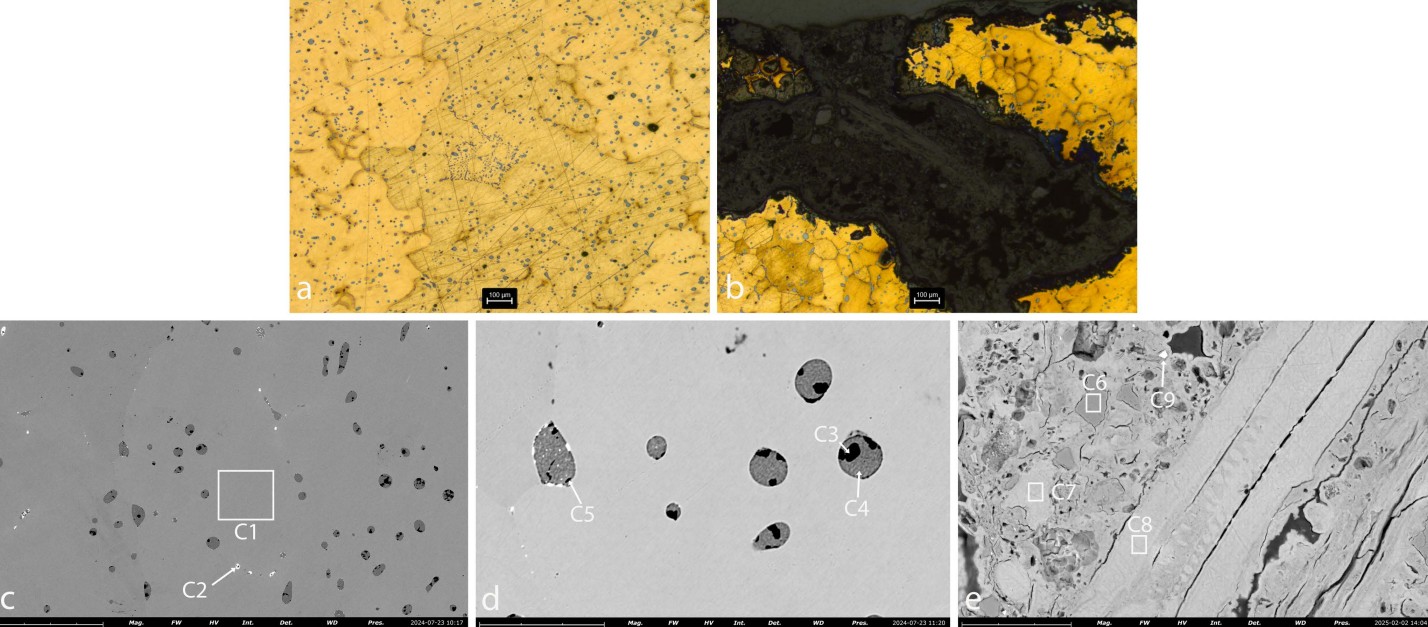

**Fig 6. Sample AM_3. a. matrix, OM, etched. b. slag, OM, etched. c-d. matrix, SEM. e. slag, SEM.** The images reveal Cu grains (spec C1) inter-spersed with globules of copper-sulfide of varied sizes (dark grey). Within the sulfur, Pb (spec C2; white), Mn-Fe (spec C3; black) and P-Fe (spec C5; black) globules are evident. Slag adheres to the copper, and dolomite minerals are detected within the slag (spec C6). Ti was measured in the slag (spec C9).

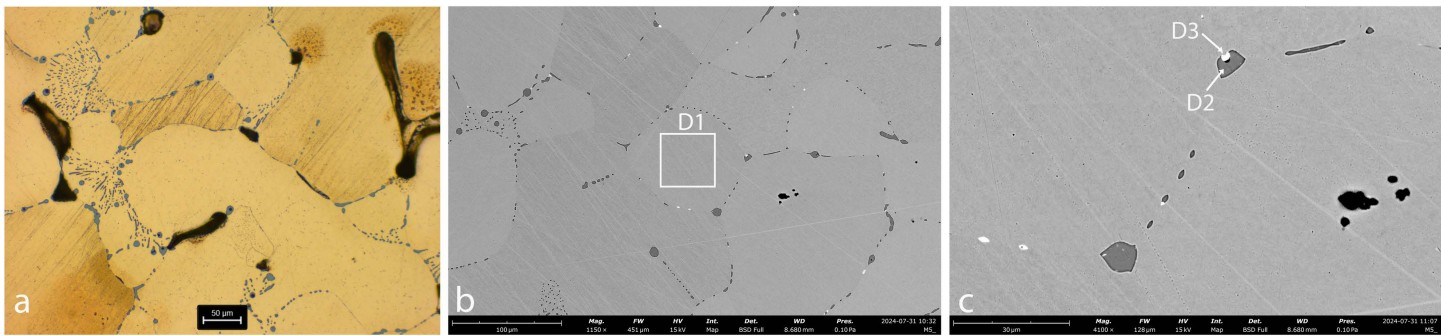

**Fig 7. Sample AM_4. a. OM, etched. b-c. SEM.** The images reveal Cu grains (spec D1) with globules of copper-sulphide (spec D2; dark-grey), mainly in the grain boundaries. Pb (white) globules are evident in grain boundaries and within Cu-S globules (spec D3).

thereby reducing the concentrations of sulfur and iron within the grains. In contrast, the remaining copper spills are characterized by an abundance of copper sulfide globules dispersed throughout the grains, and occasionally also elevated levels of iron, reflecting an incomplete separation of metal and slag during smelting.

The bronze spills, too, are of poor quality. In AM_5 and AM_6, tin was segregated to the inter-dendritic material, while the dendrites themselves consisted of nearly pure copper – an indication of especially slow cooling. AM_7 contains numerous sulfide inclusions and a high concentration of iron (17 wt.%, Table 4), suggesting poor metallurgical control. The slag fragment is especially significant, as a tin-rich prill embedded in the slag (Fig 11e) provides unequivocal evidence that the slag was the by-product of local bronze production.

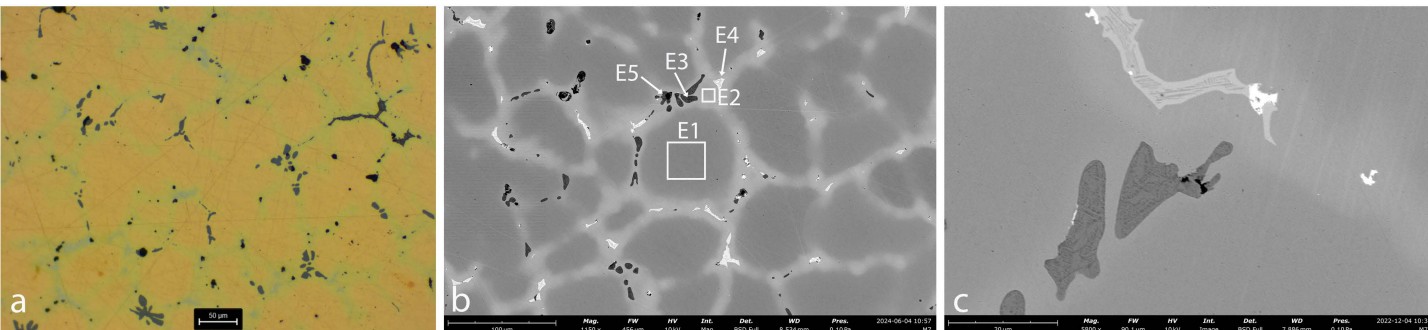

**Fig 8. Sample AM_5. a. OM, unetched. b-c. SEM.** The images reveal a tin-bronze dendritic microstructure. The dendrites contain pure Cu (spec E1), while the interdendritic material contains Cu-Sn (spec E2) interspersed with small globules of S-Fe-O (specs E3, E5; black), Sn (spec E4; light-grey) and Pb (white). The interdendritic material is also rich in As (1.2–1.5 at.%; specs E3 and E4).

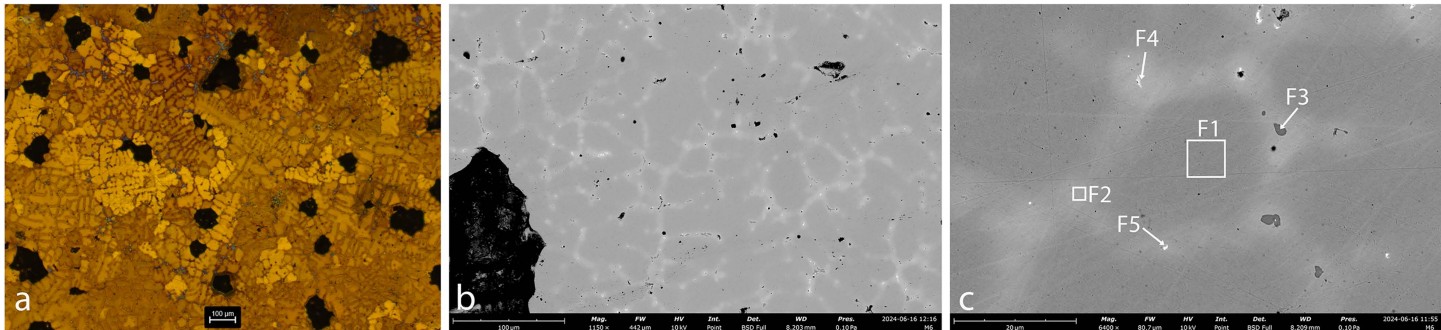

**Fig 9. Sample AM_6. a. OM, etched. b-c. SEM.** The images reveal fine Cu dendrites (spec F1). The interdendritic material contains Cu-Sn (spec F2; light-grey) interspersed with small globules of Cu-S (spec F3; dark-grey), Sn (spec F4; light-grey), Pb (spec F5; white) and Fe (black).

The term 'bronzeworking' encompasses two different metallurgical practices: the re-melting of pre-existing bronze (i.e., recycling), and 'bronzemaking', namely the primary production of bronze through the alloying of copper with tin, either in metallic or oxide form. As tin ingots are rarely found in archaeological contexts – most known examples come from underwater excavations (e.g., [79]) – direct evidence of primary bronze production is difficult to establish. Previous studies have generally interpreted bronzeworking in the southern Levant as recycling, due to a lack of systematic investigation. El-Ahwat is thus the first site in the region to yield unequivocal evidence for the primary production of bronze through alloying copper with tin.

It should be noted that the cooling rate and quality of the spills does not necessarily project on those of the produced artifacts. For example, spills may have been left to cool slowly while the production of artifacts was better controlled.

### The source of copper at el-Ahwat: Both Faynan and Timna

Lead isotope analysis (LIA) has shown that the copper used at el-Ahwat is consistent with ores from the Arabah Valley, in which two deposits are known- Timna and Faynan. Cu-S phases are abundant in many of the copper and bronze spills, also pointing to an Arabah origin [76]. Although the two ore sources overlap in Pb-isotope values, metallographic criteria have been recently suggested as a precise method of distinguishing between these sources. In particular, the presence of Fe-P-Mn-Co inclusions appear to be specific to copper from the Dolomite Limestone Shale (DLS) formations at Faynan, and are not observed in Timnah ores ([76], with some references in [5]).

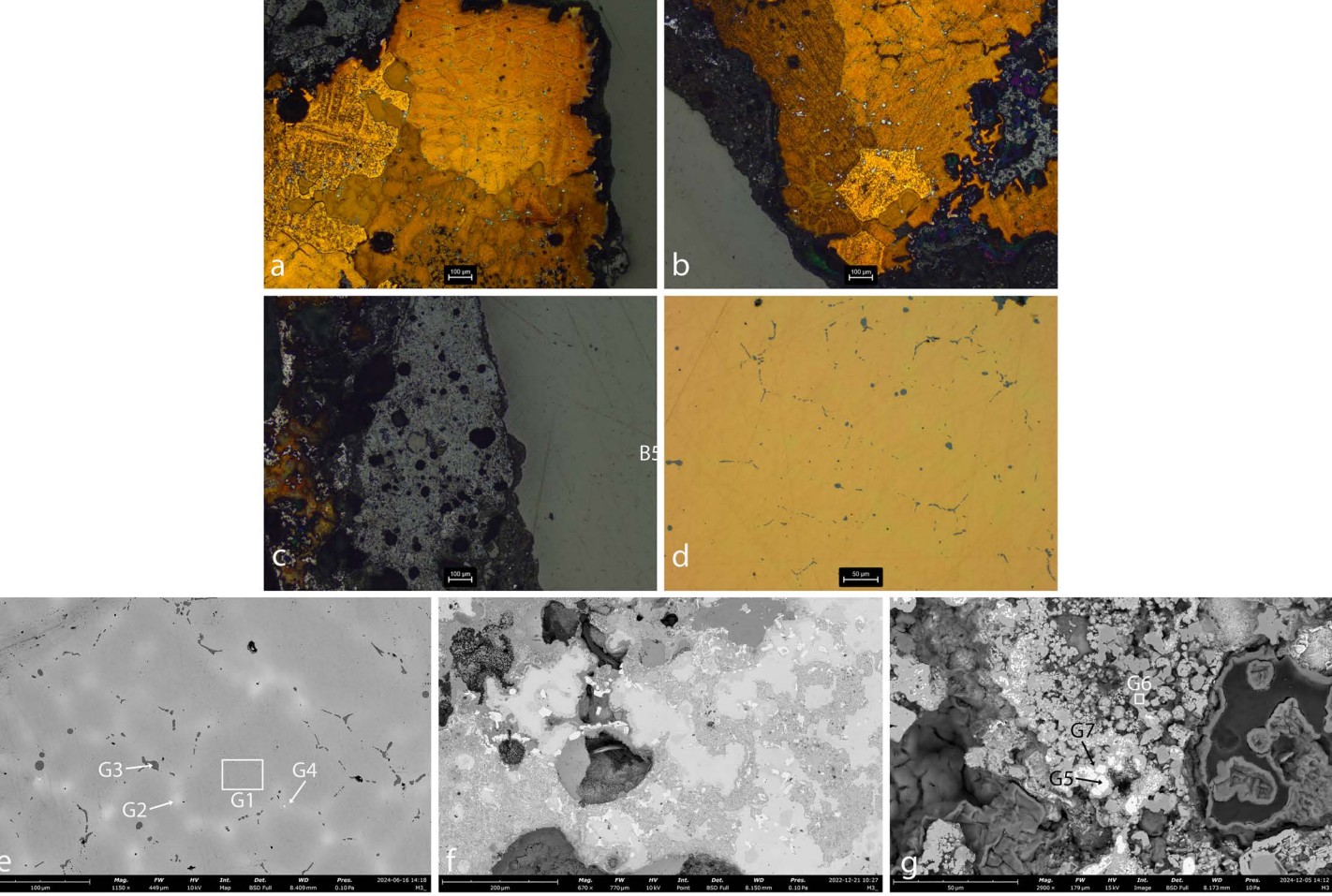

**Fig 10. Sample AM_7. a-b. matrix, OM, etched. c. slag, OM. d. matrix, OM, unetched. e. matrix, SEM. f-g. slag, SEM.** The images reveal a non-homogenous microstructure. a: Cu dendritic structures with large round inclusions in some parts; b,d.e: Cu dendrites with Fe, Sn and Mn (speck G1) in the interdendritic areas (in other parts); c,f,g: the slag observed adhering to the bronze spill is rich in Sn (specks G5, G7). The interdendritic solid contains Sn, Fe and Mn, and is interspersed with globules of copper-sulphide of different sizes (dark grey; speck G3), Sn (light grey; speck G2), Pb (white; speck G4) and Fe-O (black; speck G6).

At Timna, the presence of Mn in raw copper is only expected if Mn nodules of the Timna formation were intentionally added as a flux. Outcrops of manganese rich rocks of the dolomitic phase of the Timna formation (which is parallel to the DLS in Faynan) do occur discretely within the valley. Their use as flux was observed in slag from Timna at Site 30, dated to Iron Age IIA (late 10th – 9th c. BCE). Isotopically, the slags cluster with the manganese nodules of the Timna formation, rather than within the Amir/Avrona formation, which is more dispersed ('site 30 late' in [76]: 197, Fig 8).

In the present study, two groups of spills were distinguished: spills and slag containing P-Mn-Co inclusions (AM_2, AM_3, AM_7, AM_8), and spills that do not contain these elements (AM_1, AM_4, AM_5, AM_6). Mn and Co are closely associated (Fig 12), and are evident in both the copper and the bronze spills, suggesting that the presence of Mn, Co and/or P is not the result of alloying Cu with Sn, but rather that they originate from the copper. Since el-Ahwat pre-dates the period in which Mn was added as a flux at Timna, the chemical composition indicates that the copper originated from the

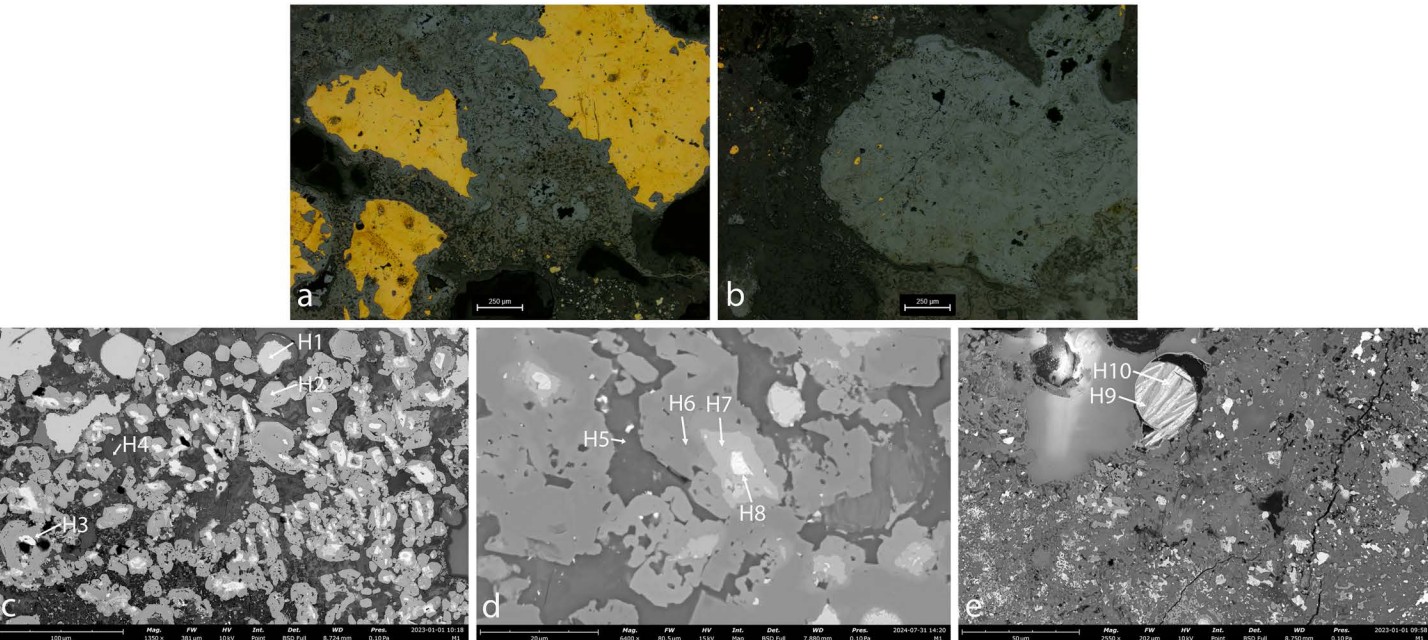

**Fig 11. Sample AM_8. a-b. OM. c.-e. SEM.** The results display copper prills (speck H1), tin oxides (specks H2, H6, H7), iron oxides (speck H5) and Sn-Cu prills (specks H8, H9), entrapped within a silica-rich slag (dark grey, specks H3, H4).

DLS P-Mn-Co rich ores [76]. Furthermore, the identification of dolomite crystals in the slag (AM_3) aligns with the geological signature of Faynan ores, which are extracted from dolomitic formations, in contrast to the sandstone geology at Timna [5].

Therefore, we suggest that the spills and slag containing Mn, Co, and often also P (AM_2, AM_3, AM_7, AM_8), contain copper from Faynan, whereas the remaining spills contain copper from Timna. While the isotopic signature of Faynan DLS is tightly clustered, none of the Mn-rich spills falls directly on the Pb-isotope values of Faynan DLS (Fig 13). This may suggest that copper from the two deposits was mixed.

These results suggest that both Faynan *and* Timna supplied copper to el-Ahwat. This conclusion should be considered in light of broader regional trends; According to the excavators, Faynan experienced intensified copper production during the Iron Age I, immediately after the Egyptian withdrawal from Canaan. This interpretation is based on radiocarbon dates from the region's largest smelting site, Khirbat en-Nahas (KEN), which covers over 10 hectares and contains more than 100 architectural features, including a large fortress and roughly 50,000–60,000 tons of slag [5,6,80,81]. The excavators date the activity at KEN to the 12th–9th centuries BCE, attributing the construction of the prominent fortress to the 10th century BCE [6]. Although some scholars have challenged this chronology, arguing that the main phase of activity at KEN began only in the 10th century BCE [82,83], evidence from a smaller site in Faynan, Khirbat al-Jariyeh, indicates that copper production was already underway during the Iron Age IB (~1050–950 BCE; [26]). Similarly, at Timna Site 30, copper production began in the 11th century BCE, with the main phase of smelting activity dated to the 10th century BCE [4]. Copper production during the Iron Age I was likely limited in quality and quantity, up until the late 10th century BCE, where a punctuated increase in production and quality is recorded, possibly attributed to the campaign of Pharaoh Shoshenq I [31,84].

The striking technological synchronicity between Timna and Faynan suggests the existence of an overarching political entity in the region as early as the 11th century BCE [84], contemporaneous with the site of el-Ahwat [67].

Table 3. Detailed chemical analysis (ICP-MS) of copper and bronze spills. Apart from copper, results are displayed in PPM.

| ID | description | SEM-EDS results | Cu wt.%* | Al | Fe | Ca | Mg | Na | K | SO4 | Sn | Pb | As | Ag | Co | Mn | Ni | Sb | Zn | Th |
|---|---|---|---|---|---|---|---|---|---|---|---|---|---|---|---|---|---|---|---|---|
| AM_1 | copper ingot | Fe in Cu matrix. boundaries contain Fe, S and Pb | 84 | n.d. | 7043 | n.d. | n.d. | n.d. | 66 | n.d. | n.d. | 4530 | 236 | n.d. | 120 | 13 | 326 | n.d. | 124 | n.d. |
| AM_2 | copper spill | Cu matrix. boundaries contain Fe, S, Pb, Mn and P. | 87 | n.d. | 15644 | n.d. | n.d. | n.d. | 81 | 6514 | n.d. | 1985 | 114 | 40 | 986 | 616 | 384 | 9 | 723 | n.d. |
| AM_3 | copper spill | Cu matrix. boundaries contain Fe, S, Pb, Mn and P. Dolomite mineral trapped in slag. | 80 | n.d. | 12270 | n.d. | n.d. | n.d. | n.d. | n.d. | 858 | 3165 | 162 | 57 | 1129 | 857 | 810 | 4 | 1787 | 8 |
| AM_4 | copper spill | Cu matrix. boundaries contain Fe, S and Pb | 104 | n.d. | n.d. | n.d. | n.d. | n.d. | n.d. | n.d. | n.d. | 688 | 56 | 60 | 84 | 23 | 281 | 8 | 176 | n.d. |
| AM_5 | bronze spill | Cu grains. Boundaries contain Fe, Sn, S and Pb | 88 | n.d. | 3083 | n.d. | n.d. | n.d. | 96 | 1698 | 5229 | 3723 | 507 | 52 | 85 | 1 | 255 | 282 | 45 | n.d. |
| AM_6 | bronze spill | Cu grains. Boundaries contain Fe, Sn, S, Pb and As | 102 | n.d. | 427 | n.d. | n.d. | n.d. | 113 | n.d. | 16273 | 3483 | 509 | 58 | 47 | 2 | 257 | 493 | 74 | n.d. |
| AM_7 | bronze spill | Cu-Sn grains. Boundaries contain Fe, Sn, S, Pb and Mn | 89 | n.d. | 17397 | n.d. | n.d. | n.d. | n.d. | n.d. | 7551 | 1629 | 137 | 59 | 239 | 101 | 259 | 14 | 237 | n.d. |
| AM_8 | bronze prill in slag | Sn-Cu prill | 71 | 6365 | 34934 | 25030 | 6897 | 1503 | 450 | 4067 | 2085 | 3262 | 293 | 22 | 314 | 1186 | 254 | 148 | 142 | n.d. |

*Most of the chemical compositions do not reach 100%, probably because of corrosion of Cu (forming CuO; [67]).

## The source of tin

The tin supply of the southern Levant depended primarily on long-distance trade networks. In recent decades, several attempts have been made to identify the sources of Late Bronze Age tin found in the region, based mainly on tin ingots recovered from underwater excavations. However, scholarly debate continues over both the methodologies and the results. Proposed source regions include Britain (Cornwall), Central Asia (Afghanistan, Iran, Tajikistan), and Anatolia [85–88]. Scholars also acknowledge that the disruptions caused by the Late Bronze Age collapse may have impacted tin availability in the southern Levant during the Early Iron Age, particularly if based on Mediterranean trade [85]. For this latter period, no tin ingots have been identified in the region, and the source of tin remains unknown.

## Edomites or Israelites? Socio-economic implications

The evidence that bronze production at el-Ahwat used copper from the Arabah challenges several longstanding socio-economic assumptions. First, it counters the idea that metals and metalworking were rare in the Central Hill Country

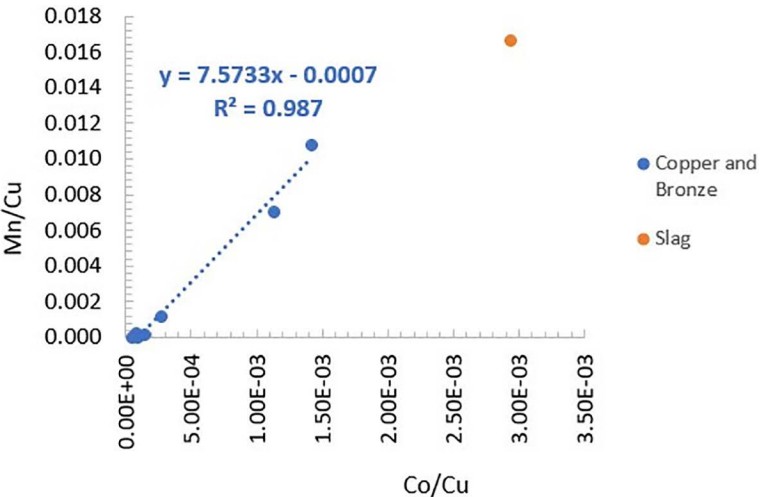

**Fig 12. Mn/Cu versus Co/Cu of the copper, bronze and slag fragments, measured by ICP-MS.**

**Table 4. Pb-isotope composition of the copper, bronze and slag fragments, with standard errors.**

|  | $^{206}$Pb/$^{204}$Pb | 2SD | $^{208}$Pb/$^{204}$Pb | 2SD | $^{207}$Pb/$^{204}$Pb | 2SD |
|---|---|---|---|---|---|---|
| **AM_1** | 17.9348 | 0.0021 | 38.0385 | 0.0073 | 15.6215 | 0.0023 |
| **AM_2** | 18.0051 | 0.0013 | 38.1308 | 0.0035 | 15.6377 | 0.0013 |
| **AM_3** | 18.0221 | 0.0007 | 38.2107 | 0.0020 | 15.6545 | 0.0008 |
| **AM_4** | 18.0923 | 0.0005 | 38.2094 | 0.0015 | 15.6417 | 0.0006 |
| **AM_5** | 18.1999 | 0.0017 | 38.3011 | 0.0067 | 15.6567 | 0.0021 |
| **AM_6** | 18.1679 | 0.0026 | 38.2917 | 0.0098 | 15.6625 | 0.0032 |
| **AM_7** | 18.0640 | 0.0013 | 38.2140 | 0.0030 | 15.6494 | 0.0010 |
| **AM_8** | 18.1915 | 0.0011 | 38.3530 | 0.0042 | 15.6646 | 0.0014 |

during Iron Age I. El-Ahwat joins other sites such as Jerusalem and Kh. Raddana, suggesting a more active metallurgical landscape.

More significantly, the data call into question the prevailing view that Iron Age I bronzeworking in the southern Levant was largely limited to domestic recycling. At el-Ahwat, copper was alloyed with tin to produce new bronze – an operation that required not only metallurgical expertise but also access to and distribution of both raw materials. This, in turn, would have required a centralized administrative system capable of coordinating resource procurement and production. The inferior quality of the copper and bronze suggests, however, a lack of experience.

While el-Ahwat provides the only direct evidence of local bronze production to date, it was likely not unique. Other sites from the same period, such as Tel Rehov and Tel Masos, have yielded bronzeworking remains with high tin concentrations (~7–20 wt.%), though these materials have yet to be studied systematically [14,54,68]. These data suggest a broader regional practice of bronze production during Iron Age I, in contrast to the more limited activity in the Late Bronze and Iron II Ages ([53]: fig II.41).

Scholars have often overlooked the fact that copper was not merely exported from the Arabah via trade routes, but was also transported inland for local bronze production. Thus, for example, Tel Masos, located in the Beersheba Valley, has been interpreted by some as a tribal chiefdom that controlled Arabah mining and directed copper transport toward

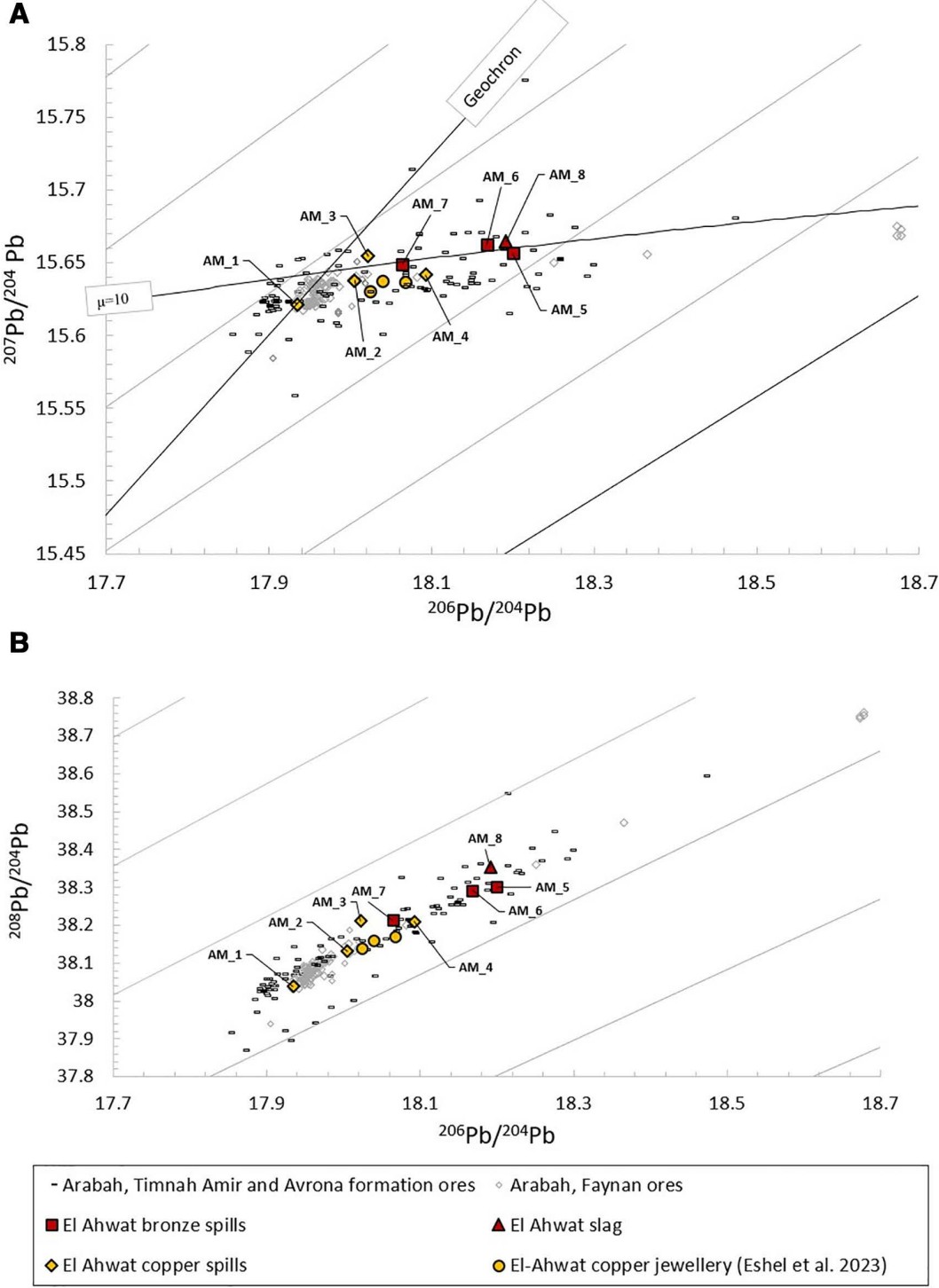

**Fig 13. Lead isotope results. A.** $^{207}Pb/^{204}Pb$ vs. $^{206}Pb/^{204}Pb$ of copper, bronze and slag from el-Ahwat; **B.** $^{208}Pb/^{204}Pb$ vs. $^{206}Pb/^{204}Pb$ of copper, bronze and slag from el-Ahwat (Table 4 and [67]). The results are plotted against the Arabah copper ores (DLS in Faynan and Amir and Evrona formations in Timna: [76–78]).

the Mediterranean and Egypt (see above). Similarly, Greek pottery from Tel Rehov in the Beth Shean Valley suggests an active trade corridor linking Phoenician coastal sites, the Jordan Valley, and Faynan ([53]; for a dissenting view see [60]). The discoveries from el-Ahwat reinforce the notion that these trade routes not only enabled the export of copper, but also facilitated the internal transportation of copper from the mines to urban bronze-producing centers.

The debate over who controlled the Faynan and Timna mines in Iron Age I – whether Edomite nomads, an emerging Edomite polity, Early Israelites, or another local power possibly based at Tel Masos – must be re-evaluated in light of the evidence from el-Ahwat. The growing number of bronzeworking centers suggests a more complex relationship between copper-producing regions in the Arabah and the administrative systems of the southern Levant, extending beyond mere trade to potential political or economic integration.

While local bronze production likely required centralized coordination, the nature of this administration remains uncertain, and it is yet to be established whether it was led by tribal elites, emerging proto-polities, or shaped by external influence. Regardless of its ethnic identity, it is clear that copper production and bronzemaking had a significant impact on Early Iron Age settlements in the Central Hill Country and across the southern Levant.

Notably, el-Ahwat and most other Central Hill Country settlements are only generally dated to Iron Age I [89,90]. More precise dating of the bronzeworking context at el Ahwat, and of additional Central Hill Country sites is required to further develop a chronological sequence and cultural interpretations.

## Conclusions

Evidence of on-site bronze production at el-Ahwat overturns long-held views that Iron Age I metallurgy was absent from the Central Hill Country, and that bronze recycling was the main practice of smiths in Iron Age I. The analytical results demonstrate that copper was alloyed with tin on-site, a process previously undocumented in this region for this period. While the quality of the bronze spills from el-Ahwat was generally low, with significant inclusions and uneven alloying, the presence of a tin-rich prill in the slag confirms that the alloying process occurred locally.

The on-site alloying of copper and tin during Iron Age I, in the aftermath of the Late Bronze Age collapse, reflects significant shifts in trade, exchange, and craft organization. Whereas bronze had previously been distributed through centralized, state-controlled networks as ingots or finished products [11], the need to alloy metals locally suggests that those systems had disintegrated, requiring communities to obtain and manage copper and tin independently. This points to the emergence of new administrations and trade networks, possibly more decentralized or informal, that continued to supply tin – a non-local resource – despite the broader breakdown of international exchange. Technologically, on-site alloying demanded metallurgical expertise, indicating that skilled craftspeople operated within or alongside small communities. This marks a departure from palace-based production, and signals a more localized, flexible model of craft specialization, potentially involving itinerant metalworkers or embedded specialists [55]. Altogether, on-site alloying demonstrates the resilience and adaptability of Iron Age societies, revealing how they restructured economic and technological practices in response to the collapse of Bronze Age political and commercial systems.

Lead isotope analysis combined with metallography indicates that the copper used at el-Ahwat originated from both Timna and Faynan Dolomite Limestone Shale (DLS) formations. The discovery of on-site bronze production at el-Ahwat indicates that a complex and organized, yet inexperienced, metallurgical industry existed in the Central Hill Country – one that required administrative coordination to acquire both copper from Faynan and Timna, and tin from as-yet unidentified sources.

These findings suggest that el-Ahwat was part of a broader socio-economic network that linked copper-producing regions in the Arabah with bronze-manufacturing centers further north. The metallurgical evidence from el-Ahwat, alongside emerging data from sites such as Tel Rehov and Tel Masos, supports the notion of a vibrant and organized bronze industry in the early Iron Age southern Levant, driven not solely by nomadic exchange but by structured systems of production, transport, and governance; that may have influenced the emergence of the Kingdoms of Israel, Judah and Edom shortly after.

## Authorization and further information on samples

All necessary permits were obtained for the study described, in compliance with all relevant regulations (Israel Antiquities Authority). The metal artefacts are temporarily housed at The Zinman Institute of Archaeology at the University of Haifa, and will be returned to the Israel Antiquities Authority after our study.

## Declaration of generative AI and AI-assisted technologies in the writing process

During the preparation of this work, the authors used Chat GPT in order to improve language and readability. After using this tool/service, the authors reviewed and edited the content as needed, and take full responsibility for the content of the publication.

## Acknowledgments

We are grateful to Amit Romano for generously sharing data and contexts from the excavation of Adam Zertal. We sincerely thank Ruth Shahak-Gross from the School of Archaeology and Maritime Cultures at the University of Haifa for enabling and contributing the SEM analysis, and Nadya Teutsch of the Israel Geological Survey for her invaluable contributions to the chemical and isotopic analyses. Their support and expertise greatly enriched this research. We are grateful to Shmuel Ariely from the Israel Institute of Materials Manufacturing Technologies, at the Technion – Israel Institute of Technology, Haifa; and to Israel Finkelstein, head of the School of Archaeology and Maritime Cultures at the University of Haifa for their insightful comments. Karan Desai assisted with the laboratory work, Aaron Lipkin skilfully produced the drone photograph, and Sapir Haad contributed to the photography and graphics. Their support is gratefully acknowledged. SEM-EDS images and analyses were conducted at the Laboratory for Sedimentary Archaeology, University of Haifa, thanks to funds provided through an equipment grant to R. Shahack-Gross by the Israel Science Foundation. We thank the reviewers and academic editor of this journal for their significant contributions which improved the manuscript. Finally, we are grateful to Elisheva Rigbi for her skillful editing.

## Author contributions

**Conceptualization:** Tzilla Eshel.

**Data curation:** Tzilla Eshel, Yoav Bornstein.

**Formal analysis:** Tzilla Eshel, Yoav Bornstein.

**Funding acquisition:** Tzilla Eshel.

**Investigation:** Tzilla Eshel, Yoav Bornstein, Gal Barmatov-Paz.

**Methodology:** Tzilla Eshel.

**Resources:** Tzilla Eshel, Shay Bar.

**Supervision:** Tzilla Eshel.

**Writing – original draft:** Tzilla Eshel.

**Writing – review & editing:** Tzilla Eshel.

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
