## [Decision Letter · Decision Letter 0]

25 May 2025

Dear Dr. Eshel,

Thank you for submitting your manuscript to PLOS ONE. After careful consideration, we feel that it has merit but does not fully meet PLOS ONE’s publication criteria as it currently stands. Therefore, we invite you to submit a revised version of the manuscript that addresses the points raised during the review process.

We look forward to receiving your revised manuscript.

Kind regards,

Joe Uziel

Academic Editor

PLOS ONE

2. In your manuscript, please provide additional information regarding the specimens used in your study. Ensure that you have reported human remain specimen numbers and complete repository information, including museum name and geographic location.

For more information on PLOS ONE's requirements for paleontology and archeology research, see https://journals.plos.org/plosone/s/submission-guidelines#loc-paleontology-and-archaeology-research.

“This research was supported by THE ISRAEL SCIENCE FOUNDATION (grant No. 1032/23, awarded to T.E.)”

4. We note that Figures 1 and 2 in your submission contain [map/satellite] images which may be copyrighted. All PLOS content is published under the Creative Commons Attribution License (CC BY 4.0), which means that the manuscript, images, and Supporting Information files will be freely available online, and any third party is permitted to access, download, copy, distribute, and use these materials in any way, even commercially, with proper attribution. For these reasons, we cannot publish previously copyrighted maps or satellite images created using proprietary data, such as Google software (Google Maps, Street View, and Earth). For more information, see our copyright guidelines: http://journals.plos.org/plosone/s/licenses-and-copyright.

a. You may seek permission from the original copyright holder of Figures 1 and 2 to publish the content specifically under the CC BY 4.0 license. 

Reviewers' comments:

Reviewer's Responses to Questions

**Comments to the Author**

1. Is the manuscript technically sound, and do the data support the conclusions?

Reviewer #1: Yes

Reviewer #2: Yes

2. Has the statistical analysis been performed appropriately and rigorously?

Reviewer #1: Yes

Reviewer #2: Yes

3. Have the authors made all data underlying the findings in their manuscript fully available?

Reviewer #1: Yes

Reviewer #2: Yes

4. Is the manuscript presented in an intelligible fashion and written in standard English?

Reviewer #1: Yes

Reviewer #2: Yes

Reviewer #1: General Assessment

This is a well-structured and thoroughly researched paper that makes a significant and original contribution to the archaeology of early Iron Age metallurgy in the Southern Levant. The article presents new archaeometallurgical evidence from the site of el-Ahwat, arguing convincingly for local bronze production during Iron Age I, based on metallography, chemical analysis (ICP-MS), and lead isotope analysis (LIA).

The link to Arabah copper ores (Faynan and Timna) is important and timely, and the combination of scientific data and broader historical interpretation is particularly valuable. Yet, the attempt to trace copper provenance to both Timna and Faynan using lead isotopes and microstructural features is ambitious and intriguing. However, given the known isotopic overlap between ores from these two regions, and the emerging but still debated metallographic criteria, this identification should be presented with a degree of caution. While the authors address this overlap, the conclusion of a mixed or dual origin could benefit from more critical engagement with the methodological limitations.

Strengths

1. Original Contribution:

The paper presents the first unequivocal evidence of local bronze production (not recycling) in the Iron Age I Southern Levant, filling a significant gap in the scholarly literature.

2. Multidisciplinary Approach:

The authors combine multiple lines of evidence—optical microscopy, SEM-EDS, ICP-MS, and LIA—to build a strong case. Their discussion is data-rich and methodologically rigorous.

3. Clear and Comprehensive Presentation:

The article is well written and organized, with clear tables, detailed methodology, and well-argued conclusions. The inclusion of Figures, Tables, and Appendices is appropriate and supports the findings.

4. Regional and Historical Significance:

The findings have broader implications for understanding the economic and political networks in the Iron Age Southern Levant and offer new perspectives on the role of inland highland communities in early metallurgical production.

Points for Improvement / Clarification

1. Copper Provenance – Timna vs. Faynan:

While the integration of LIA and metallography is commendable, there remains inherent ambiguity in distinguishing between Timna and Faynan copper due to their overlapping isotopic signatures. The use of micro-inclusions (Mn, Co, P) as distinguishing markers for Faynan's DLS ores is promising, but still under development in the field. The authors may consider tempering the strength of their attribution and explicitly acknowledging the tentative nature of such identifications, even when the evidence is suggestive.

2. Discussion of Tin Sources:

The paper notes that the source of tin remains unidentified. While this is understandable, it would be beneficial to briefly discuss possible routes or regions of origin for tin in this period (e.g., Anatolia, Central Asia, or re-export via Mediterranean trade), even if speculative.

3. Elaboration on Administrative Structures:

The suggestion that local bronze production required centralized coordination is persuasive. However, the authors could expand on what form this administration may have taken—tribal elites, proto-polities, or external influence.

4. Chronological Range:

Although the general dating of el-Ahwat to Iron Age I is provided, more precise dating of the bronzeworking context (e.g., specific strata, C14 ranges if available) would strengthen the chronological argument, especially in light of the discussions of Timna and Faynan.

Recommendation: Accept with Minor Revisions

I strongly support the publication of this article in PLOS ONE pending minor revisions, especially a more cautious framing of copper source attribution to Timna and Faynan. The manuscript is otherwise of excellent quality and makes a meaningful contribution to the study of early Iron Age metallurgy in the Levant.

Reviewer #2: I would like the authors to include and address all the points below:

Line 36 - Benefiters – make it “beneficiaries” Suggested sentence: One of the beneficiaries of this dramatic change were the Iron Age societies inhabiting the copper ore districts in the Arabah valley that separates modern Israel and Jordan.

Line 38 – add reference –(Levy and Najjar 2007)

Line 39 – insert ‘Israel’ to make “…at Timna on the western side of the Wadi Arabah in Israel (Fig. 1),

Line 54 – “…copper used at destination sites cannot yet be determined” should be - copper used at destination sites is difficult to determine…

Line 63 – “… add ‘ancient’ to make ‘ancient Edom’

Line 65 - Excavators have proposed that semi-nomadic Edomite groups engaged in herding and limited copper production occupied Faynan and Timna. DELETE – ‘limited’

Line 86 – add two references - (Galili et al. 2020; Yasur-Landau et al. 2021)

Line 131 – delete ‘never’ replace with ‘not’

Line 137 – define what a metal spill is.

Line 339 – delete ‘or’ replace with ‘and’ to make ‘Timna and Faynan’

Line 340 - While the two ore sources overlap in Pb-isotope values, metallographic criteria have been recently suggested as a precise method of distinguishing these sources.’

Note: Long ago, Shalev showed that Faynan ore could be distinguished from Timna ores based on metallographic analyses, as did Hauptmann. See and add the following references: (Hauptmann 2007; Shalev and Northover 1987)

Line 358 – add reference - (Levy, Najjar, Higham, et al. 2014)

Line 364 – The increase in copper production in the 10th c. BCE was not gradual. It was a fast, punctuated change. This point needs to be highlighted. See - (Ben-Yosef et al. 2019)

Lines 374 – 375 - Chronologically, el-Ahwat is dated to the Iron Age I, ~1200–950 BCE, predating Shoshenk I (Eshel et al. 2023), in accordance with the low quality of the copper found onsite.

Not clear what the authors mean by low quality of the copper . What is this based on? Do the authors mean ‘limited quantity’ of copper in the 10th c. BCE at Khirbat en-Nahas? In discussing the pre-10th c BCE at Kh. En-Nahas, one has to factor in the very small excavation exposures (due to the fact that the pre-10th c. BCE material is found near the base of the slag mounds, over 6 meters in depth making it difficult to expose).

Conclusion -

Line 412 – “results demonstrate that copper was alloyed with tin on-site, a process previously undocumented in this region for this period.

Elaborate what on-site alloying implies for trade and exchange, as well as craft specialization during the Iron I considering the post Late Bronze Age collapse.

Line 426 – “emergence of the Kingdom of Israel and Judah shortly after.”

These conclusions also apply to the Edomite Kingdom. It should be included here.

References to add or correct -

Ben-Yosef, E., B. Liss, O.A. Yagel, O. Tirosh, M. Najjar, and Levy T.E.

2019 Ancient technology and punctuated change: Detecting the emergence of the Edomite Kingdom in the Southern Levant. PLoS ONE 14(9: e0221967):https://doi.org/10.1371/journal.pone.0221967.

Galili, Ehud, Baruch Rosen, Mina Weinstein Evron, Israel Hershkovitz, Vered Eshed, and Liora Kolska Horwitz

2020 Israel: Submerged Prehistoric Sites and Settlements on the Mediterranean Coastline—the Current State of the Art. In The Archaeology of Europe’s Drowned Landscapes, edited by Geoff Bailey, Nena Galanidou, Hans Peeters, Hauke Jöns, and Moritz Mennenga, pp. 443-481. Springer International Publishing, Cham.

Hauptmann, A

2007 The Archaeo-metallurgy of Copper - Evidence from Faynan, Jordan. Springer, New York.

Levy, T.E., and M. Najjar

2007 Ancient Metal Production and Social Change in Southern Jordan: The Edom Lowlands Regional Archaeology Project and Hope for a UNESCO World Heritage Site in Faynan. In Crossing Jordan - North American Contributions to the Archaeology of Jordan, edited by T.E. Levy, M. Daviau, R.W. Younker, and M. Shaer, pp. 97-105. Equinox, London.

Levy, T.E., M. Najjar, and E. Ben-Yosef

2014 New Insights into the Iron Age Archaeology of Edom, Southern Jordan - Surveys, Excavations and Research from the Edom Lowlands Regional Archaeology Project (ELRAP). Two Volumes. Cotsen Institute of Archaeology Press UCLA, Los Angeles.

Levy, T.E., M. Najjar, T. Higham, Y. Arbel, A. Muniz, E. Ben-Yosef, N.G. Smith, M. Beherec, A.D. Gidding, I.W.N. Jones, D. Freses, and M. Robinson

2014 Excavations at Khirbat en-Nahas 2002-2009: An Iron Age Copper Production Center in the Lowlands of Edom. In New Insights into the Iron Age Archaeology of Edom, Southern Jordan - Surveys, Excavations and Research from the Edom Lowlands Regional Archaeology Project (ELRAP), edited by T.E. Levy, M. Najjar, and E. Ben-Yosef, pp. 89 - 245. UCLA Cotsen Institute of Archaeology Press, Los Angeles.

Shalev, S., and J. P. Northover

1987 Chalcolithic metal and metalworking from Shiqmim. In Shiqmim I - Studies Concerning Chalcolithic Socieites in the Northern Negev Desert, Israel (1982 - 1984), edited by T.E. Levy, pp. 357-371; 683-689. BAR International Series 356, Oxford.

Yasur-Landau, Assaf, Gilad Shtienberg, Gil Gambash, Giorgio Spada, Daniele Melini, Ehud Arkin-Shalev, Anthony Tamberino, Jack Reese, Thomas E. Levy, and Dorit Sivan

2021 New relative sea-level (RSL) indications from the Eastern Mediterranean: Middle Bronze Age to the Roman period (~3800–1800 y BP) archaeological constructions at Dor, the Carmel coast, Israel. PLOS ONE 16(6):e0251870.

**Do you want your identity to be public for this peer review?** For information about this choice, including consent withdrawal, please see our Privacy Policy

Reviewer #1: No

Reviewer #2: **Yes: ** Thomas E. Levy

---

## [Author Response · Author response to Decision Letter 1]

11 Jul 2025

07/07/2025

Reply to review of the paper PONE-D-25-24485 “First Evidence of Bronze Production in the Iron Age I Southern Levant - A Direct Link to the Arabah Copper Polity”

Dear Dr. Uziel,

Please find my replies below in blue to all the comments made by you and by the reviewers.

The edits following the reviewer comments and additional changes are highlighted in the main manuscript. I appreciate the time and effort that you put into this process, and believe that the comments have substantially improved the paper. As you will see, the comments were considered and the manuscript was modified accordingly.

Sincerely yours,

Tzilla Eshel

This was reviewed by a professional editor. Style requirements were considered and implemented throughout the manuscript.

Ensure that you have reported museum name and geographic location. If permits were required, please ensure that you have provided details for all permits that were obtained, including the full name of the issuing authority, and add the following statement: 'All necessary permits were obtained for the described study, which complied with all relevant regulations.' If no permits were required, please include the following statement: 'No permits were required for the described study, which complied with all relevant regulations.'

Added, see lines 462–463.

included, see lines 465–468.

We note that Figures 1 and 2 in your submission contain [map/satellite] images which may be copyrighted.

Figure 1: A new figure was prepared by a graphic designer, attached.

Figure 2: A written permission was obtained from the photographer. It is not a reprint but an original ariel photo from a drone.

Reference list was reviewed.

A list of added cited papers (in un-edited format):

Reference (and text) of a proposed iron furnace at El Awhat

Winter, Y. (2012). A Furnace for the Processing of Iron. In: Zertal, A., ed. El-Aḥwat: A Fortified Site from the Early Iron Age near Naḥal Iron (CHANE 24). Leiden and Boston: 381–385.

Reference (and text) on the debated chronology of Khirbet en-Nahas, Faynan:

Tebes, J. M. (2022). A reassessment of the chronology of the iron age site of Khirbet en-Nahas, Southern Jordan. Palestine exploration quarterly, 154(2), 113-140.

Finkelstein, I. (2005). Khirbet en-Nahas, Edom and biblical history. Tel Aviv, 32(1), 119-125.

Reference (and text) regarding the dating of the Central Hill Country settlement wave during the Iron Age I, following the reviewer’s comments:

Gadot, Y. (2019). The Iron I Settlement Wave in the Samaria Highlands and Its Connection with the Urban Centers. Near Eastern Archaeology 82: 32–41

Sharon, I., Gilboa, A., Jull, A.T. and Boaretto, E. (2007). Report on the First Stage of the Iron Age Dating Project in Israel: Supporting a Low Chronology. Radiocarbon 49: 1–46.

Reference (and text) on tin, following the reviewer’s comment:

Berger, D., Soles, J. S., Giumlia-Mair, A. R., Brügmann, G., Galili, E., Lockhoff, N., & Pernicka, E. (2019). Isotope systematics and chemical composition of tin ingots from Mochlos (Crete) and other Late Bronze Age sites in the eastern Mediterranean Sea: An ultimate key to tin provenance?. PloS one, 14(6), e0218326.

Berger, D., Kaniuth, K., Boroffka, N., Brügmann, G., Kraus, S., Lutz, J., ... & Pernicka, E. (2023). The rise of bronze in Central Asia: new evidence for the origin of Bronze Age tin and copper from multi-analytical research. Frontiers in Earth Science, 11, 1224873.

Powell, W., Johnson, M., Pulak, C., Yener, K. A., Mathur, R., Bankoff, H. A., ... & Galili, E. (2021). From peaks to ports: Insights into tin provenance, production, and distribution from adapted applications of lead isotopic analysis of the Uluburun tin ingots. Journal of Archaeological Science, 134, 105455.

Powell, W., Frachetti, M., Pulak, C., Bankoff, H. A., Barjamovic, G., Johnson, M., ... & Yener, K. A. (2022). Tin from Uluburun shipwreck shows small-scale commodity exchange fueled continental tin supply across Late Bronze Age Eurasia. Science Advances, 8(48), eabq3766.

Additional references recommended by the reviewer and found relevant by the author:

Galili, E., Rosen, B., Evron, M. W., Hershkovitz, I., Eshed, V., & Horwitz, L. K. (2020). Israel: Submerged prehistoric sites and settlements on the Mediterranean coastline—The current state of the art. In Bailey, G., Galanidou, N., Peeters, H., Jöns, H. and Mennenga, M. (Eds.) The Archaeology of Europe’s Drowned Landscapes (pp. 443-481). Springer International Publishing, Cham.

Levy, T.E., and Najjar, M. (2007). Ancient Metal Production and Social Change in Southern Jordan: The Edom Lowlands Regional Archaeology Project and Hope for a UNESCO World Heritage Site in Faynan. In T.E. Levy, M. Daviau, R.W. Younker, and M. Shaer (Eds.). Crossing Jordan - North American Contributions to the Archaeology of Jordan (pp. 97-105). Equinox, London.

Levy, T.E., Najjar, M., Higham, T., Arbel, Y., Muniz, A., Ben-Yosef, E., Smith N.G., Beherec, M., Gidding, A.D., Jones, I.W.N., Freses, D. and Robinson, M. (2014b). Excavations at Khirbat en-Nahas 2002-2009: An Iron Age Copper Production Center in the Lowlands of Edom. In New Insights into the Iron Age Archaeology of Edom, Southern Jordan - Surveys, Excavations and Research from the Edom Lowlands Regional Archaeology Project (ELRAP), T.E. Levy, M. Najjar, and E. Ben-Yosef (Eds.). (pp. 89 – 245). UCLA Cotsen Institute of Archaeology Press, Los Angeles.

Yasur-Landau, A., Shtienberg, G., Gambash, G., Spada, G., Melini, D., Arkin-Shalev, E., ... & Sivan, D. (2021). New relative sea-level (RSL) indications from the Eastern Mediterranean: Middle Bronze Age to the Roman period (~ 3800–1800 y BP) archaeological constructions at Dor, the Carmel coast, Israel. PloS one, 16(6), e0251870.

Reviewers' comments:

Reviewer #1: General Assessment

The link to Arabah copper ores (Faynan and Timna) is important and timely, and the combination of scientific data and broader historical interpretation is particularly valuable. Yet, the attempt to trace copper provenance to both Timna and Faynan using lead isotopes and microstructural features is ambitious and intriguing. However, given the known isotopic overlap between ores from these two regions, and the emerging but still debated metallographic criteria, this identification should be presented with a degree of caution. While the authors address this overlap, the conclusion of a mixed or dual origin could benefit from more critical engagement with the methodological limitations.

I am not aware of a debate regarding the most recent and comprehensive attempt of Bode et al. 2023 to trace copper provenance to both Timna and Faynan using lead isotopes and microstructural features. Nonetheless, due to the concerns raised by the reviewer, a degree of caution was added throughout the discussion (see e.g., lines 355, 357).

I also added some explanation regarding the possibility of Mn in Timna ores in later periods, see lines 339– 344:

“At Timna, the presence of Mn in raw copper is only expected if Mn nodules of the Timna formation were intentionally added as a flux. Outcrops of manganese rich rocks of the dolomitic phase of the Timna formation (which is parallel to the DLS in Faynan) do occur discretely within the valley. Their use as flux was observed in slag from Timna at Site 30, dated to Iron Age IIA (late 10th – 9th c. BCE). Isotopically, the slags cluster with the manganese nodules of the Timna Formation, rather than within the Amir/Avrona formation, which is more dispersed (‘site 30 late’ in [76]: 197, Fig. 8).”

Discussion of Tin Sources:

The paper notes that the source of tin remains unidentified. While this is understandable, it would be beneficial to briefly discuss possible routes or regions of origin for tin in this period (e.g., Anatolia, Central Asia, or re-export via Mediterranean trade), even if speculative.

This was added to the main text, see lines 372–380.

Elaboration on Administrative Structures:

The suggestion that local bronze production required centralized coordination is persuasive. However, the authors could expand on what form this administration may have taken—tribal elites, proto-polities, or external influence.

I don’t think the results can elaborate on this directly, but I added a discussion of possibilities. See lines 410–413.

4. Chronological Range:

Although the general dating of el-Ahwat to Iron Age I is provided, more precise dating of the bronzeworking context (e.g., specific strata, C14 ranges if available) would strengthen the chronological argument, especially in light of the discussions of Timna and Faynan.

Bronzeworking was particularly prominent in Iron Age IB, as mentioned in lines 73–75 with references (for specific strata and chronology). Carbon-14 dates for the central hill country are lacking. this was added and explained in lines 414–416.

Reviewer #2: I would like the authors to include and address all the points below:

Line 36 - Benefiters – make it “beneficiaries” Suggested sentence: One of the beneficiaries of this dramatic change were the Iron Age societies inhabiting the copper ore districts in the Arabah valley that separates modern Israel and Jordan.

corrected, lines 35-36.

Line 38 – add reference –(Levy and Najjar 2007)

added, line 38.

Line 39 – insert ‘Israel’ to make “…at Timna on the western side of the Wadi Arabah in Israel (Fig. 1)

added, line 39.

Line 54 – “…copper used at destination sites cannot yet be determined” should be - copper used at destination sites is difficult to determine…

corrected, line 50.

Line 63 – “… add ‘ancient’ to make ‘ancient Edom’

added, line 56.

Line 65 - Excavators have proposed that semi-nomadic Edomite groups engaged in herding and limited copper production occupied Faynan and Timna. DELETE – ‘limited’

deleted, line 58.

Line 86 – add two references - (Galili et al. 2020; Yasur-Landau et al. 2021)

added, line 73.

Line 131 – delete ‘never’ replace with ‘not’

deleted

Line 137 – define what a metal spill is.

added, lines 120–123.

Line 339 – delete ‘or’ replace with ‘and’ to make ‘Timna and Faynan’

replaced

Line 340 - While the two ore sources overlap in Pb-isotope values, metallographic criteria have been recently suggested as a precise method of distinguishing these sources.’

Note: Long ago, Shalev showed that Faynan ore could be distinguished from Timna ores based on metallographic analyses, as did Hauptmann. See and add the following references: (Hauptmann 2007; Shalev and Northover 1987)

Bode et al. 2023 stands out in its large volume of analyses. Hauptmann 2007 was added in lines 337–338. I also added some explanation regarding the possibility of Mn in Timna ores (see above).

Line 358 – add reference - (Levy, Najjar, Higham, et al. 2014)

added

Line 364 – The increase in copper production in the 10th c. BCE was not gradual. It was a fast, punctuated change. This point needs to be highlighted. See - (Ben-Yosef et al. 2019)

rephrased.

lines 368–370: “Copper production during the Iron Age I was likely limited in quality and quantity, up until the late 10th century BCE, where a punctuated increase in production and quality is recorded, possibly attributed to the campaign of Pharaoh Shoshenq I [31, 84].”

Lines 374 – 375 - Chronologically, el-Ahwat is dated to the Iron Age I, ~1200–950 BCE, predating Shoshenk I (Eshel et al. 2023), in accordance with the low quality of the copper found onsite.

Rephrased, see comment above.

Not clear what the authors mean by low quality of the copper . What is this based on? Do the authors mean ‘limited quantity’ of copper in the 10th c. BCE at Khirbat en-Nahas? In discussing the pre-10th c BCE at Kh. En-Nahas, one has to factor in the very small excavation exposures (due to the fact that the pre-10th c. BCE material is found near the base of the slag mounds, over 6 meters in depth making it difficult to expose).

I added a clarification in lines 328–330: “It should be noted that the cooling rate and quality of the spills does not necessarily project on those of the produced artifacts. For example, spills may have been left to cool slowly while the production of artifacts was better controlled”.

Also, I added a chronological discussion regarding the excavations at Faynan and Timna (lines 357–368):

“These results suggest that both Faynan and Timna supplied copper to el-Ahwat. This conclusion should be considered in light of broader regional trends; According to the excavators, Faynan experienced intensified copper production during the Iron Age I, immediately after the Egyptian withdrawal from Canaan. This interpretation is based on radiocarbon dates from the region’s largest smelting site, Khirbat en-Nahas (KEN), which covers over 10 hectares and contains more than 100 architectural features, including a large fortress and roughly 50,000–60,000 tons of slag [5, 6, 80, 81]. The excavators date the activity at KEN to the 12th–9th centuries BCE, attributing the construction of the prominent fortress to the 10th century BCE [6]. Although some scholars have challenged this chronology, arguing that the main phase of activity at KEN began only in the 10th century BCE [82, 83], evidence from a smaller site in Faynan, Khirbat al-Jariyeh (KAJ), indicates that copper production was already underway during the Iron Age IB (~1050–950 BCE; [26]). Similarly, at Timna Site 30, copper production began in the 11th century BCE, with the main phase of smelting activity dated to the 10th century BCE [4].”

Conclusion -

Line 412 – “results demonstrate that copper was alloyed with tin on-site, a process previously undocumented in this region for this period.

Elaborate what on-site alloying implies for trade and exchange, as well as craft specialization during the Iron I considering the post Late Bronze Age collapse.

Added, lines 426–437: “The on-site alloying of copper and tin during Iron Age I, in the aftermath of the Late Bronze Age collapse, reflects significant shifts in trade, exchange, and craft organization. Whereas bronze had previously been distributed through centralized, state-controlled networks as ingots or finished [11], the need to alloy metals locally suggests that those systems had disintegrated, requiring communities to obtain and manage copper and tin independently. This points to the emergence of new administrations and trade networks, possibly more decentralized or informal, that continued to supply tin - a non-local resource - despite the broader breakdown of international exchange. Technologically, on-site alloying demanded metallurgical expertise, indicating that skilled craftspeople operated within or alongside small communities. This marks a departure from palace-based production, and signals a more localized, flexible model of craft specialization, potentially involving itinerant metalworkers or embedded specialists [55]. Altogether, on-site alloying demonstrates the resilience and adaptability of Iron Age societies, revealing how they restructured economic and technological practices in response to the collapse of Bronze Age political and commercial systems.”

Line 426 – “emergence of the Kingdom of Israel and Judah shortly after.”

These conclusions also apply to the Edomite Kingdom. It should be included here.

added, line 448.

References to add or correct - Most of them added, see above.

Ben-Yosef, E., B. Liss, O.A. Yagel, O. Tirosh, M. Najjar, and Levy T.E.

2019 Ancient technology and punctuated change:

---

## [Editor Report · Decision Letter 1]

13 Jul 2025

First Evidence of Bronze Production in the Iron Age I Southern Levant - A Direct Link to the Arabah Copper Polity

PONE-D-25-24485R1

Dear Dr. Eshel,

We’re pleased to inform you that your manuscript has been judged scientifically suitable for publication and will be formally accepted for publication once it meets all outstanding technical requirements.

Kind regards,

Joe Uziel

Academic Editor

PLOS ONE
---

## [Editor Report · Acceptance letter]

PONE-D-25-24485R1

PLOS ONE

Dear Dr. Eshel,

I'm pleased to inform you that your manuscript has been deemed suitable for publication in PLOS ONE. Congratulations! Your manuscript is now being handed over to our production team.

Kind regards,

on behalf of

Dr. Joe Uziel

Academic Editor

PLOS ONE